# Single cell preparations of *Mycobacterium tuberculosis* damage the mycobacterial envelope and disrupt macrophage interactions

Ekansh Mittal[1,2†], Andrew T Roth[3†], Anushree Seth[4], Srikanth Singamaneni[4,5], Wandy Beatty[2], Jennifer A Philips[1,2]*

[1]Division of Infectious Diseases, Department of Medicine, Washington University School of Medicine, St Louis, United States; [2]Department of Molecular Microbiology, Washington University School of Medicine, St Louis, United States; [3]Division of Pulmonary and Critical Care Medicine, Department of Medicine, Washington University School of Medicine, St Louis, United States; [4]Department of Mechanical Engineering and Materials Science, Institute of Materials Science and Engineering, Washington University in St. Louis, St Louis, United States; [5]Siteman Cancer Center, Washington University, St. Louis, United States

*For correspondence:
philips.j.a@wustl.edu

†These authors contributed equally to this work

**Abstract** For decades, investigators have studied the interaction of *Mycobacterium tuberculosis* (Mtb) with macrophages, which serve as a major cellular niche for the bacilli. Because Mtb are prone to aggregation, investigators rely on varied methods to disaggregate the bacteria for these studies. Here, we examined the impact of routinely used preparation methods on bacterial cell envelope integrity, macrophage inflammatory responses, and intracellular Mtb survival. We found that both gentle sonication and filtering damaged the mycobacterial cell envelope and markedly impacted the outcome of infections in mouse bone marrow-derived macrophages. Unexpectedly, sonicated bacilli were hyperinflammatory, eliciting dramatically higher TLR2-dependent gene expression and elevated secretion of IL-1β and TNF-α. Despite evoking enhanced inflammatory responses, sonicated bacilli replicated normally in macrophages. In contrast, Mtb that had been passed through a filter induced little inflammatory response, and they were attenuated in macrophages. Previous work suggests that the mycobacterial cell envelope lipid, phthiocerol dimycocerosate (PDIM), dampens macrophage inflammatory responses to Mtb. However, we found that the impact of PDIM depended on the method used to prepare Mtb. In conclusion, widely used methodologies to disaggregate Mtb may introduce experimental artifacts in Mtb-host interaction studies, including alteration of host inflammatory signaling, intracellular bacterial survival, and interpretation of bacterial mutants.

## Editor's evaluation

This important study examines how the method used to prepare *Mycobacterium tuberculosis* bacilli for use in experimental infection models affects outcomes. The authors provide compelling evidence indicating that the method of preparation affects the bacterial cell wall and substantially influences both host responses and infection outcomes in models of TB infection. The data will be useful for interpreting published TB literature and for future studies probing *M. tuberculosis* virulence. The work will be of interest to tuberculosis researchers and microbiologists in general.

## Introduction

A fundamental feature of the pathogenesis of *Mycobacterium tuberculosis* (Mtb), the etiologic agent of tuberculosis (TB), is its ability to survive and grow in host macrophages. For more than five decades, many laboratories have investigated how Mtb interacts with and modulates the function of macrophages. Mtb is characterized by a 'waxy' coat, which confers its distinctive acid-fast staining properties. The complex cell envelope is important for pathogenesis and also allows Mtb to withstand adverse conditions (*Dulberger et al., 2020*). The mycobacterial envelope consists of a plasma membrane, peptidoglycan-arabinogalactan layer, outer membrane, and capsular layer (*Dulberger et al., 2020*). The outermost layers of the envelope are crucial in host-pathogen interactions given that they are directly able to interact with host cells. The capsule is composed primarily of a loose matrix of neutral polysaccharides (*Kalscheuer et al., 2019*), while the outer membrane is composed of long-chain mycolic fatty acids that are free, attached to trehalose, or covalently attached to the underlying arabinogalactan-peptidoglycan layer. The outer membrane also contains a complex array of unique lipids. Many of these lipids are bioactive; they can intercalate into host membranes, alter inflammatory signaling, disrupt phagosome maturation, and promote mycobacterial virulence (*Cambier et al., 2020*; *Lerner et al., 2018*; *Quigley et al., 2017*). Some outer membrane lipids, including phthiocerol dimycocerosate (PDIM), phenolic glycolipids, and sulfoglycolipids, are thought to act as antagonists of pathogen recognition receptors (PRRs) or to shield underlying pathogen associated molecular patterns (PAMPs) to prevent them from activating PRRs (*Blanc et al., 2017*; *Cambier et al., 2014*; *Reed et al., 2004*). Thus, the integrity of the envelope is crucial for host interactions and bacterial virulence.

Given the importance of Mtb-macrophage interactions, a mainstay of the experimental approach of many laboratories is the use of in vitro cultured Mtb to infect myeloid cells. However, the tendency of Mtb to form bacterial clumps has long presented an obstacle to these experiments, which depend on using precise and reproducible amounts of bacteria (*Wells, 1946*). For this reason, low concentrations of detergents are commonly added to culture media, but this does not fully resolve the problem. Therefore, additional measures are routinely taken to generate single cell suspensions, including sonicating, syringing, centrifuging, filtering, vortexing with glass beads, or some combination of these procedures. The use of these techniques varies widely across different laboratories, the methodology used is not always reported, and there has been little consideration as to how these techniques impact experimental outcomes.

We are interested in how mycobacterial protein and lipid effectors modulate macrophage responses. Sometimes our results differed from published data, leading us to question whether the method of preparing the bacilli explained the differences. However, there was minimal literature into how dispersing mycobacterial clumps impacts the envelope and host-pathogen interactions. Previous studies demonstrated that use of detergent and agitation can release capsular constituents (*Lemassu et al., 1996*; *Sani et al., 2010*). In addition, it was reported that Mtb that had been sonicated for 90 s were better able to bind to macrophages, and the bacterial envelope appeared uneven and bulging on transmission electron microscopy (*Stokes et al., 2004*). Another study showed that passing bacterial cultures through 5 µm pore filters improved reproducibility of high-throughput antibacterial drug screening compared to vortexing, but the impact on host-pathogen interactions was not assessed (*Cheng et al., 2014*). Given the lack of published studies addressing our concern, we compared three routinely used methods of preparing single cell suspensions of Mtb: low-speed spin, gentle sonication followed by low-speed spin, or filtration through a 5 µm filter. We found that the method of bacterial preparation had a marked impact on intracellular bacterial viability, the global transcriptional pattern of infected cells, macrophage secretion of key innate immune mediators, and the ultrastructure of the bacterial cell envelope. Finally, when comparing an Mtb mutant that lacks PDIM to WT bacilli, we found that the method of preparation had a substantial impact on the inflammatory response to the mutant bacilli.

## Results

### Macrophage transcriptional responses to Mtb depend on the method of bacterial preparation

To investigate whether the method of dispersing bacterial cultures impacts host responses, we examined gene expression profiles of bonemarrow-derived macrophages (BMDMs) that were uninfected or infected with WT Mtb (H37Rv strain) that were prepared either by passing through a 5 µm filter (5µmF) or by brief sonication (so). The sonicated samples underwent three 10 second cycles in a water bath sonicator followed by a low-speed spin (sp), as described in Materials and methods, and are designated so/sp. Bacilli prepared by the two methods were added to BMDMs at a multiplicity of infection (MOI) of 5, washed to remove extracellular bacteria after 4 hr, and processed for RNA-seq 72 hr post-infection (hpi). We found that 536 genes were differentially expressed between uninfected cells and both of the infected samples (adjusted p-value ≤0.01; fold change ≥|2|). Surprisingly, however, there were even more genes that were differentially expressed uniquely in macrophages infected by only one of the two bacterial preparations. A total of 902 differentially expressed genes (DEGs) were unique when we compared uninfected with so/sp-infected macrophages, while 122 genes were uniquely differentially expressed in response to 5µmF bacteria (*Figure 1A*; *Figure 1—figure supplement 1A*; *Supplementary file 1*). When we compared the DEGs in BMDM infected with so/sp-Mtb to those infected with the 5µmF-preparations, there were 732 DEGs (*Figure 1A–B*). These included genes encoding important host defense molecules, including *Il6*, *Nos2*, and *Il1b*, which were markedly higher in so/sp-infected BMDMs compared to 5µmF-infected BMDMs (*Figure 1B*).

In order to further analyze the transcriptional differences, we used Gene Set Enrichment Analysis (GSEA) to query our expression data against hallmark gene sets from the Molecular Signatures Database (*Liberzon et al., 2015*). We found that 10 hallmark gene sets were significantly enriched in so/sp preparations relative to 5µmF-infected BMDMs (p≤0.01; FDR ≤0.01) (*Figure 1C–D*, *Figure 1—figure supplement 1B*). The sonicated bacilli elicited a significant enrichment of gene sets that included TNFA signaling via NFKB, inflammatory response, MTORC1 signaling, glycolysis, xenobiotic metabolism, and IL6 JAK STAT3 signaling (*Figure 1C*, *Figure 1—figure supplement 1B*). In contrast, 1 hallmark gene set was significantly enriched in 5µmF- relative to so/sp-infected BMDMs (E2F targets) (*Figure 1—figure supplement 1C*). Visualization of transcriptional data from the hallmark gene set 'inflammatory response' showed a distinct gene expression pattern in response to so/sp versus 5µmF bacteria (*Figure 1D*). Overall, the macrophages infected with the so/sp bacilli displayed a more robust pro-inflammatory phenotype, whereas the 5µmF-infected macrophages were enriched in pro-replication pathways. In addition to the 72 hpi timepoint used for RNA-seq, we found that infection with so/sp Mtb elicited significantly higher levels of expression of *Il1b*, *Nos2*, *Il6*, and *Tnf* at 6 and 24 hpi by qPCR compared with 5µmF preparations, with the greatest difference seen early in infection (*Figure 1E–F*). Strikingly, while the so/sp bacilli markedly upregulated inflammatory gene expression, there was minimal difference between 5µmF-infected and uninfected macrophage 6 hpi and 24 hpi. In these experiments, we used 5 µm filters with hydrophilic polyethersulfone (PES) membranes, which are composed of aryl-$SO_2$-aryl subunits. We considered the possibility that the chemical backbone and hydrophilic nature of the filter might be altering Mtb, but we had similar findings when we used hydrophobic polytetrafluoroethylene (PTFE) filters (*Figure 1—figure supplement 2*). In conclusion, the transcriptional response of BMDMs to Mtb infection was markedly different depending on the method of bacterial preparation.

### Sonication increases the inflammatory impact of Mtb

Given the dramatic difference between so/sp and 5µmF bacilli, it was important to assess which one more accurately reflects unperturbed Mtb. However, if we were to use Mtb directly from a liquid culture, it would not be possible to establish that we are using similar numbers of bacilli compared to the other preparations given the propensity to clump. Therefore, we used a low-speed spin preparation. This was the same procedure applied to the so/sp sample, but the sonication step was omitted. Specifically, liquid cultures were centrifuged at 206 x g for 10 min, after which the supernatant was removed and centrifuged at 132 x g for 8 min, and the final supernatant was used to infect BMDMs. We found that macrophages infected with the spin (sp) sample had an intermediate phenotype between so/sp and 5µmF samples (*Figure 2A*), eliciting significantly less *Il1b*, *Il6*, *Nos2*, and *Tnf* expression than the so/sp samples. To determine whether the impact of preparation method was specific to

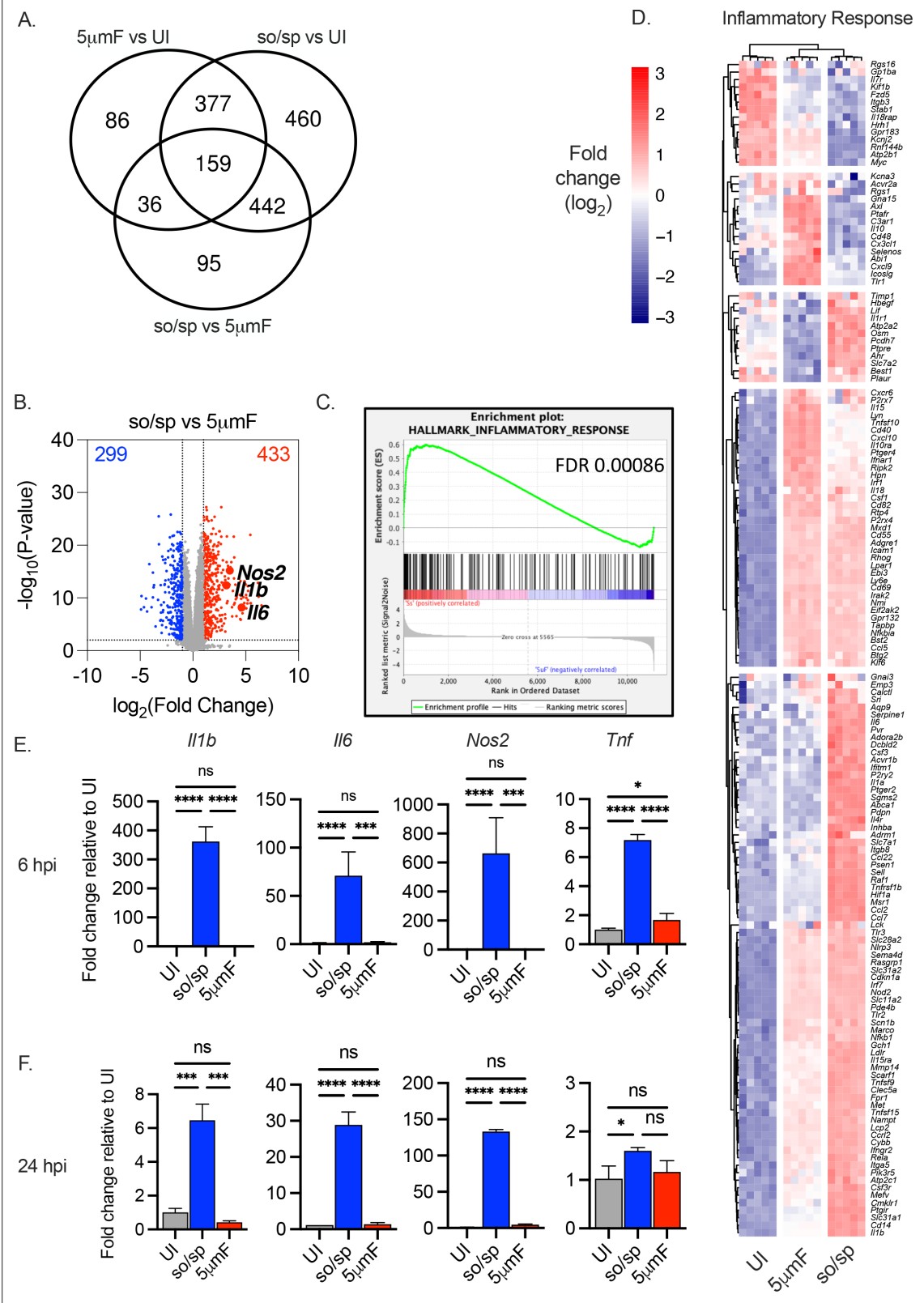

**Figure 1.** Cell preparation methods of Mtb impact macrophage responses. (**A–D**) BMDMs were uninfected or infected with Mtb prepared by sonication and spin (so/sp) or filtration (5μmF) at an MOI of 5 and analyzed 72 hpi by RNA-seq (n=5 per condition). (**A**) Venn diagram illustrates the number of DEGs between samples. (**B**) Volcano plot shows genes differentially expressed in BMDMs infected with so/sp versus 5μmF Mtb. DEGs exhibiting an adjusted p-value of ≤0.01 and a linear fold change ≥2.00 (red) or ≤2.00 (blue) are indicated. (**C–D**) GSEA identified hallmark gene sets that were

*Figure 1 continued*

significantly enriched in so/sp- versus 5µmF-infected BMDMs (p≤0.01; FDR ≤0.01). Representative enrichment plot (**C**) and corresponding heat map (**D**) for the gene set 'inflammatory response'. Expression values in heatmap were generated using log2 normalized CPM for each gene. (**E and F**) qPCR was performed on uninfected BMDMs or BMDMs infected with so/sp- or 5µmF-prepared Mtb 6 (**E**) and 24 (**F**) hpi using an MOI of 10. Data are shown as fold change in gene expression relative to uninfected BMDMs. Data shown are mean +/-SD from one representative experiment with three biological replicates per group and two technical replicates per sample. qPCR experiments were performed at least three independent times. Statistical significance was determined with one-way ANOVA using Tukey's multiple comparisons test. (**A–F**) Error bars indicate mean +/-SD. ns not significant; *p<0.05; ***<0.001; ****<0.0001.

The online version of this article includes the following figure supplement(s) for figure 1:

**Figure supplement 1.** Infection-induced changes in macrophage gene expression depend on whether Mtb are sonicated or filtered.

**Figure supplement 2.** Filtered Mtb are uninflammatory irrespective of filter type.

H37Rv, we tested two additional Mtb strains: HN878, a W-Beijing lineage strain that was isolated in a TB outbreak in Houston in the 1990s, and Erdman, a strain that is commonly used is laboratory studies. We found that the method of preparation had a similar impact on macrophage inflammatory responses for these strains as for H37Rv (*Figure 2—figure supplement 1*). A variety of Mtb PAMPs have been shown to activate TLR2 (*Hinman et al., 2021*). In order to establish whether the so/sp samples were activating TLR2-dependent pathways, we infected BMDMs from *Tlr2*−/− mice. We found that expression of *Il1b*, *Il6*, *Nos2*, and *Tnf* were significantly reduced in TLR2 KO BMDMs in response to so/sp Mtb relative to WT BMDMs (*Figure 2B*). This was also true for the induction observed in response to spin preparations.

We wondered if the 5µmF bacilli contained a factor that inhibited macrophage gene expression or if they were just less proinflammatory. To address this, we mixed so/sp and 5µmF bacilli together in equal proportions and assayed gene expression by qPCR. We found that the mixed samples were still inflammatory, arguing against a potent inhibitory factor coming from the filtered preparation (*Figure 2—figure supplement 2A*). In addition, if we prepared bacteria by first sonicating and then using a 5µmF (so/5µmF), the expression changes resembled so/sp infection, with marked upregulation of *Il1b*, *Il6*, *Nos2*, and *Tnf* (*Figure 2A*). We considered the possibility that the different inflammatory responses might be a result of different degrees of aggregation of the bacilli in each preparation. To visualize the bacteria, we infected BMDMs with GFP-expressing Mtb prepared by the various methods and examined them by fluorescence microscopy (*Figure 2C–D*). We quantified whether the visualized bacteria were single/doublets (1-2), small (3-6), or large (>6) clumps. For all of the preparations, more than 75% of the bacterial occurrences were single/doublets. The 5µmF and so/5µmF preparations had slightly more single/doublets and slightly fewer clumps than the other samples (*Figure 2C–D*). Since the clumpiness of so/sp and sp samples were similar, aggregation status did not explain the hyper-inflammatory nature of the so/sp samples. In addition, when the sonicated sample was filtered (so/5µmF), it had few clumps, and the bacilli still induced high levels of *Il1b*, *Il6*, *Nos2*, and *Tnf* (*Figure 2A–D*). To investigate whether the response to sonicated bacteria was due to soluble factors released from the bacilli, we passed the so/sp sample through a 0.2 µm filter to remove bacteria. We treated macrophages with equal volumes of the sterile filtrate or the unfiltered so/sp sample and analyzed subsequent gene expression. In support of extra-bacterial components contributing to the inflammatory gene expression, the expression of *Il1b*, *Nos2*, *Il6*, and *Tnf* were all significantly increased in response to the sterile filtrate prepared from the so/sp bacteria compared to uninfected BMDMs (*Figure 2E*). In contrast, there was no difference in expression of these genes in the sterile filtrate of sp or 5µmF bacteria relative to uninfected BMDMs (*Figure 2E*, *Figure 2—figure supplement 2B*). To conclude, compared to bacteria prepared by a low-speed spin or 5µmF, bacilli that were sonicated induced substantially higher TLR2-dependent transcriptional responses in macrophages, independent of their aggregation status and due in part to soluble mediators.

To determine whether the changes in gene expression resulted in altered cytokine secretion, we used the FluoroDOT assay to evaluate secretion of TNF-α. This approach uses plasmon-enhanced fluorescent nanoparticles called plasmonic fluors to visualize protein secretion by microscopy (*Figure 3—figure supplement 1*). This allowed us to examine secretion of TNF-α at an early time point after infection and with single cell resolution (*Seth et al., 2022*). Similar to the transcriptional data, the sonicated preparations elicited the most TNF-α secretion followed by the sp and 5µmF preparations (*Figure 3A–B*). We confirmed these findings by measuring TNF-α by enzyme linked

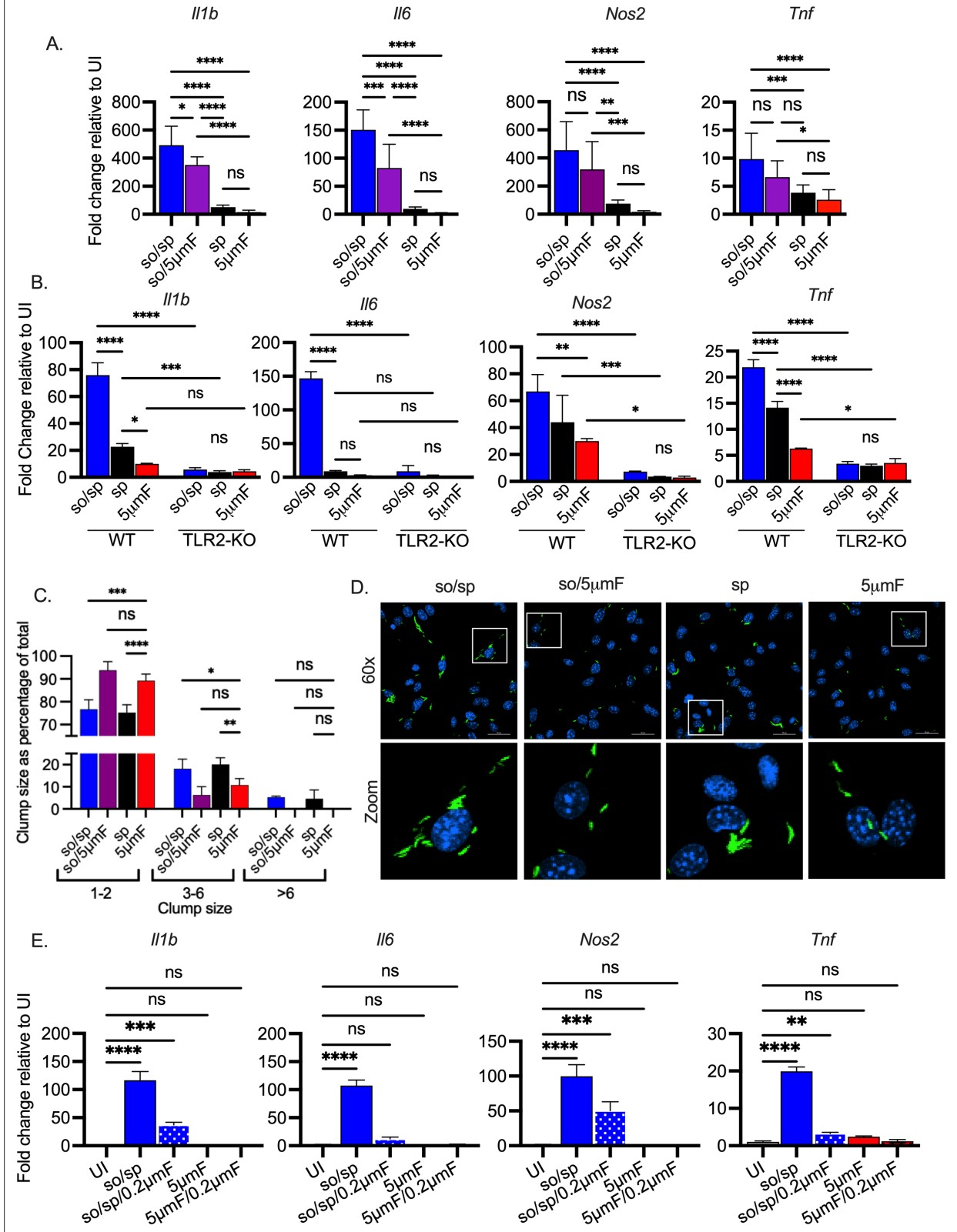

**Figure 2.** Sonicated bacteria induce high TLR2-dependent inflammatory responses. (**A**) BMDMs were uninfected or infected with different preparations of Mtb as indicated at an MOI of 10 and analyzed by qPCR 6 hpi. Data are presented as fold changes in gene expression relative to uninfected BMDMs. Data are combined from two to three experiments, each with three biological replicates per group and two technical replicates per sample. Statistical significance was determined with one-way ANOVA using Tukey's multiple comparisons test. (**B**) WT or *Tlr2*[-/-] BMDMs were uninfected or infected with

*Figure 2 continued on next page*

*Figure 2 continued*

different preparations of Mtb as indicated for 6 hr at an MOI of 10 and analyzed by qPCR. Data are presented as fold change in gene expression relative to uninfected BMDMs of the same mouse genotype. Data are representative of three experiments, each with three biological replicates per group and two technical replicates per sample. Statistical significance was determined with two-way ANOVA using Tukey's multiple comparisons test. (**C**) BMDMs were infected with GFP-expressing bacteria (MOI 5) and visualized using immunofluorescence at 4 hpi. Bacteria were quantified and classified as single/doublets (1-2), small (3-6), or large (>6) clumps and quantified for each preparation. At least 100 bacterial occurrences of each preparation method were analyzed. Two-way ANOVA with Dunnett's multiple comparisons test was used to assess statistical significance within each batch relative to the given 5µmF quantitation. (**D**) Representative fluorescence microscopy images of BMDMs (nuclei stained with DAPI) infected with GFP-expressing Mtb used in (**C**). Images are maximum-intensity projections. Boxed areas in the merged image are shown in higher magnification in the bottom panel. (**E**) BMDMs were untreated, infected with the indicated bacterial preparations at MOI 10, or treated with the sterile filtrate from different preparations for 6 hr and analyzed by qPCR. Data are presented as fold changes in gene expression relative to untreated BMDMs. Data are representative of 3 experiments, each with three biological replicates per group and two technical replicates per sample. Statistical significance was determined for each group relative to untreated BMDMs with one-way ANOVA using Dunnett's multiple comparisons test. (**A–B, D**) Error bars indicate mean +/-SD. ns not significant; *p<0.05; **<0.01; ***<0.001; ****<0.0001.

The online version of this article includes the following figure supplement(s) for figure 2:

**Figure supplement 1.** Sonicated HN878 and Erdman strains induce high TLR2-dependent inflammatory responses.

**Figure supplement 2.** Filtered bacteria do not strongly inhibit response from sonicated bacteria and only the sterile filtrate of so/sp Mtb induces gene expression in BMDMs.

---

immunosorbent assay (ELISA; *Figure 3C*). We also evaluated IL-1β secretion using ELISA and found that the so/sp preparation elicited increased secretion of IL-1β (*Figure 3C*). Interestingly, we found that when infected by the sonicated samples, most of the macrophages, both infected as well as uninfected bystanders in the same well, secreted TNF-α. In contrast, infection with the sp or 5µmF Mtb resulted in only infected cells secreting TNF-α(*Figure 3A*). In addition, so/sp samples that had been sterilized by passage through a 0.2 µm filter elicited significantly more TNF-α secretion than sterilized 5µmF samples (*Figure 3D–E*). This is consistent with the observation that extra-bacterial components in the sonicated preparation contribute to inflammatory gene expression.

## Filtered Mtb are attenuated in BMDMs

Given that the different preparations generated pronounced differences in macrophage gene expression and cytokine secretion, we hypothesized that they would also exhibit differences in intracellular viability. We infected BMDMs with Mtb prepared by the different methods. To ensure that a similar MOI was used for each bacterial preparation, we plated the input used for the infection. We had to use 1.5-times more filtered (so/5µmF and 5µmF) bacilli based on $OD_{600}$ to achieve the same number of viable bacteria. After infection, the intracellular bacilli were enumerated at 4 hpi and 3 and 5 days post-infection (dpi). Mtb that were prepared by so/sp or sp grew in macrophages significantly better than those that were filtered (so/5µmF and 5µmF; *Figure 4A*). Increasing the MOI of the filtered bacteria to 20 or 40 did not overcome the intracellular growth defect (*Figure 4B*). The differences in intracellular growth were not explained by differences in macrophage viability (*Figure 4C*). We verified that the filtered Mtb were still viable, as they grew indistinguishably from other preparations when they were inoculated in liquid culture (*Figure 4D*). Similar to our findings with H37Rv, filtered HN878 and Erdman were also attenuated in BMDMs compared to those prepared by so/sp or sp (*Figure 4E–F*). Thus, filtered Mtb appeared to be both less inflammatory and impaired in their ability to counter the antimicrobial properties of BMDMs.

## Sonication and filtering affect the bacterial cell wall

To determine if there were structural differences between the sonicated, spun, and filtered Mtb, we used transmission electron microscopy (TEM). We first generated ultrathin cross-sections of bacteria to visualize the ultrastructure of the cell envelope (*Figure 5A–C*). In bacteria prepared with low-speed spin, we could distinguish the structural layers of the cell envelope that have been previously described: the innermost phospholipid bilayer, followed by electron-dense peptidoglycan and arabinogalactan layers, a translucent mycobacterial outer membrane, and an outermost carbohydrate-rich capsular layer (*Figure 5B*). In bacteria prepared with low-speed spin and/or sonication, each of these distinct layers were apparent (*Figure 5A–B*). In the 5µmF-prepared bacteria, the phospholipid bilayer was seen, surrounded by an electron dense layer, but there appeared to be loss of the capsular

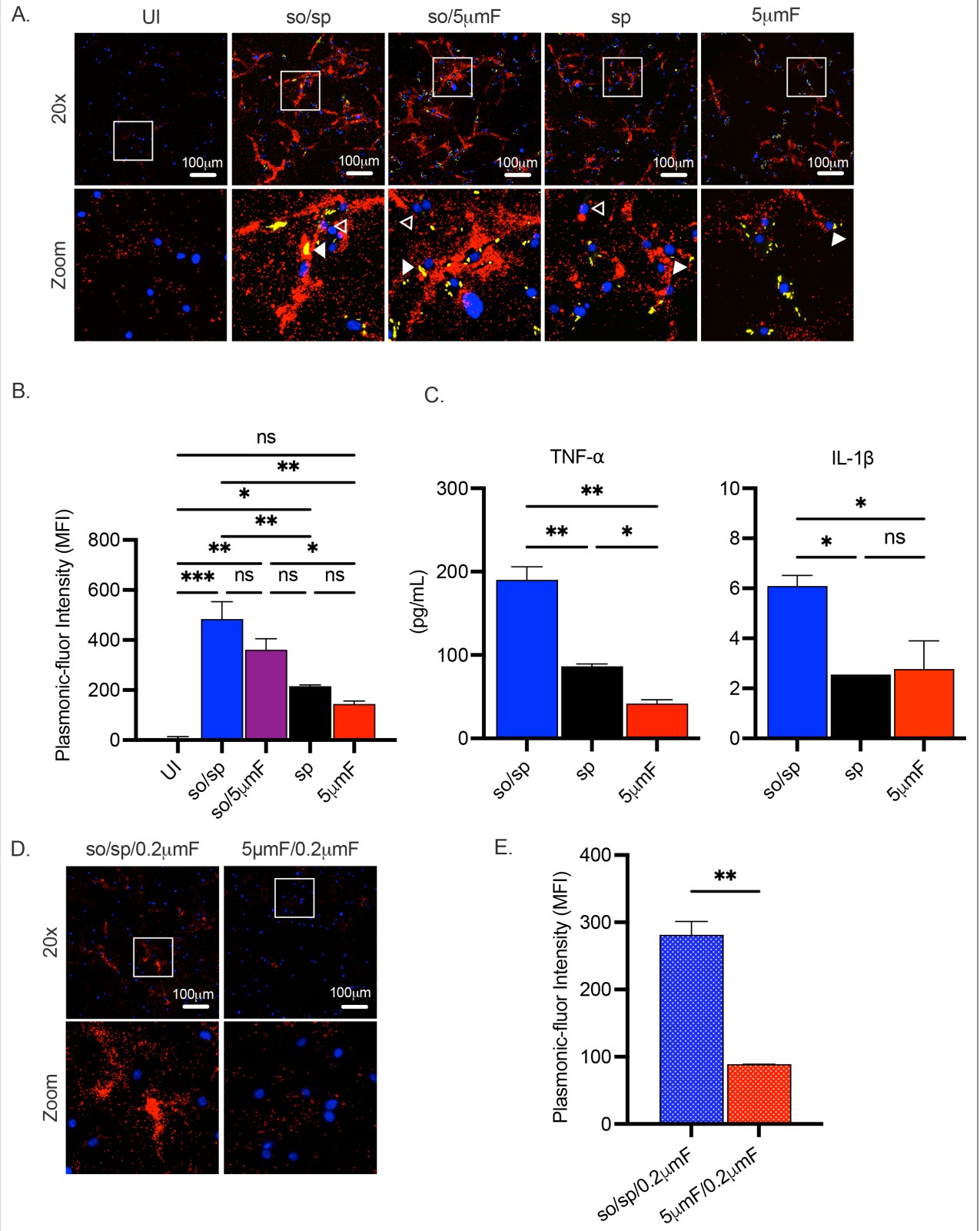

**Figure 3.** Sonicated bacteria elicit elevated TNF-α and IL-1β secretion. (**A**) Using the FluoroDOT assay, BMDMs were grown on a glass bottom plate that was coated with TNF-α capture antibody, infected at an MOI of 10 with H37Rv-GFP prepared by the indicated method, and examined by epifluorescence microscopy (20 X) 6 hpi. Images show Plasmonic-fluor 650 (red), Mtb (GFP), and DAPI (blue). Boxed areas in the image are enlarged in the bottom images. Secretion from infected BMDMs or uninfected bystander cells are highlighted by open or closed white arrowheads, respectively.

*Figure 3 continued on next page*

*Figure 3 continued*

(**B**) Data show the quantification of the mean fluorescence intensity (MFI) of the plasmonic-fluor in the entire well from each different condition shown in (**A**), with statistical significance determined with one-way ANOVA using Tukey's multiple comparisons test. (**C**) IL-1β and TNF-α were measured 24 hpi in the culture supernatant of uninfected or Mtb-infected BMDMs (MOI 10) by ELISA. Data shown are mean +/-SD from one representative experiment with three biological replicates per group and two technical replicates per sample. Significance was determined using one-way ANOVA with Tukeys' multiple comparisons test. (**D**) Using the FluoroDOT assay, BMDMs grown on a glass bottom plate that was coated with TNF-α capture antibody were exposed to the sterile filtrate of bacterial single cell suspension prepared by either so/sp or 5μmF and examined by epifluorescence microscopy (20 X) 6 hpi. Images show Plasmonic-fluor 650 (red) and DAPI (blue). Boxed areas in the image are enlarged in the bottom images. (**E**) Data show the quantification of the mean fluorescence intensity (MFI) of the plasmonic-fluor in the entire well from each condition shown in D, with statistical significance determined using an unpaired T test. (**A–E**) Error bars indicate mean +/-SD. ns not significant; *p<0.05; **<0.01; ***<0.001.

The online version of this article includes the following figure supplement(s) for figure 3:

**Figure supplement 1.** Schematic representation of the FluoroDOT assay.

layer and potentially the mycomembrane as well (*Figure 5C*). While TEM of ultrathin cross-sections provided excellent resolution of the cell wall, it was also subject to artifact introduced by drying and fracturing of the bacteria required in this technique. This made it difficult to know how representative the well-preserved bacilli were in terms of the total population. Therefore, we also visualized bacilli by adsorption to a copper grid followed by 1% uranyl acetate staining, a simple technique which minimized artifact (*Figure 5D–F*). Uranyl acetate is a common negative stain used for TEM that can bind to capsular polysaccharides (*Stukalov et al., 2008*), and it created an electron dense halo around the bacteria. We quantified the width of the electron dense halo on individual bacilli and found that it was significantly thinner on bacteria prepared with the 5μmF compared to so/sp and sp (*Figure 5G*). This suggests a different chemical composition of the outermost layer of the filtered bacteria and was consistent with the differences noted in the TEM. In addition, the samples from 5μmF-treated bacteria had substantial extracellular debris, which may be damaged fragments from the outer layers of the envelope. Finally, more dead bacteria were noted in the 5μmF sample as evidenced by penetration of the dark staining uranyl acetate into the cells (*Figure 5F*), which may explain why we had to use 1.5-times more 5μmF bacilli (based upon optical density) to achieve the same number of viable bacteria. Using this technique, we also observed that the so/sp bacteria, but not sp or 5μmF bacteria, had prominent round protuberances that were approximately 0.2–1 μM in diameter present on the outer surface of the bacteria or, less frequently, in the culture media (*Figure 5D*). To conclude, both sonicated and filtered preparations had evidence of distinct types of damage to the envelope on TEM that were not apparent in the samples which had been prepared by centrifugation alone.

## The interpretation of the role of PDIM in inflammatory responses depends upon preparation method

PDIM is a multifunctional virulence lipid that is present in the envelope of members of the Mtb complex as well as closely related *Mycobacterium marinum*. Along with the ESX-1 type VII secretion system, PDIM facilitates phagosomal escape of Mtb, a crucial event that allows the bacteria to gain access to the cytosol, subvert cell death pathways, and promote extracellular spread (*Augenstreich et al., 2017*; *Barczak et al., 2017*; *Cox et al., 1999*; *Lerner et al., 2018*; *Osman et al., 2020*; *Quigley et al., 2017*). In addition, PDIM contributes to the low permeability of the mycobacterial envelope, alters the host's initial innate immune response, and may physically shield mycobacterial PAMPs or interfere with their activation of PRRs (*Astarie-Dequeker et al., 2009*; *Camacho et al., 2001*; *Cambier et al., 2014*; *Murry et al., 2009*; *Rousseau et al., 2004*; *Siméone et al., 2007*). To determine whether PDIM dampens inflammatory signaling, we used a strain with a deletion in *ppsD*, which results the in the absence of PDIM (*Barczak et al., 2017*). When we examined macrophage gene expression after infection with Δ*ppsD* by qPCR, we found that expression of *Il1b*, *Il6*, *Nos2*, and *Tnf* was significantly increased compared to infection with WT Mtb, consistent with the idea that PDIM reduces inflammatory signaling (*Figure 6A*). However, this was only significant and reproducible in the sonicated sample; there was little difference between Δ*ppsD* and WT Mtb if they were prepared by sp or 5μmF. We had similar findings when we used the FluoroDOT assay to examine TNF-α secretion (*Figure 6B–C*). The Δ*ppsD* mutant reproducibly elicited more TNF−α secretion than WT Mtb, but only if the sample was sonicated. Interestingly, when we examined the Δ*ppsD* mutant by TEM, we found that the Δ*ppsD* mutant lacked the dark halo that was seen in so/sp and sp samples of WT Mtb; the

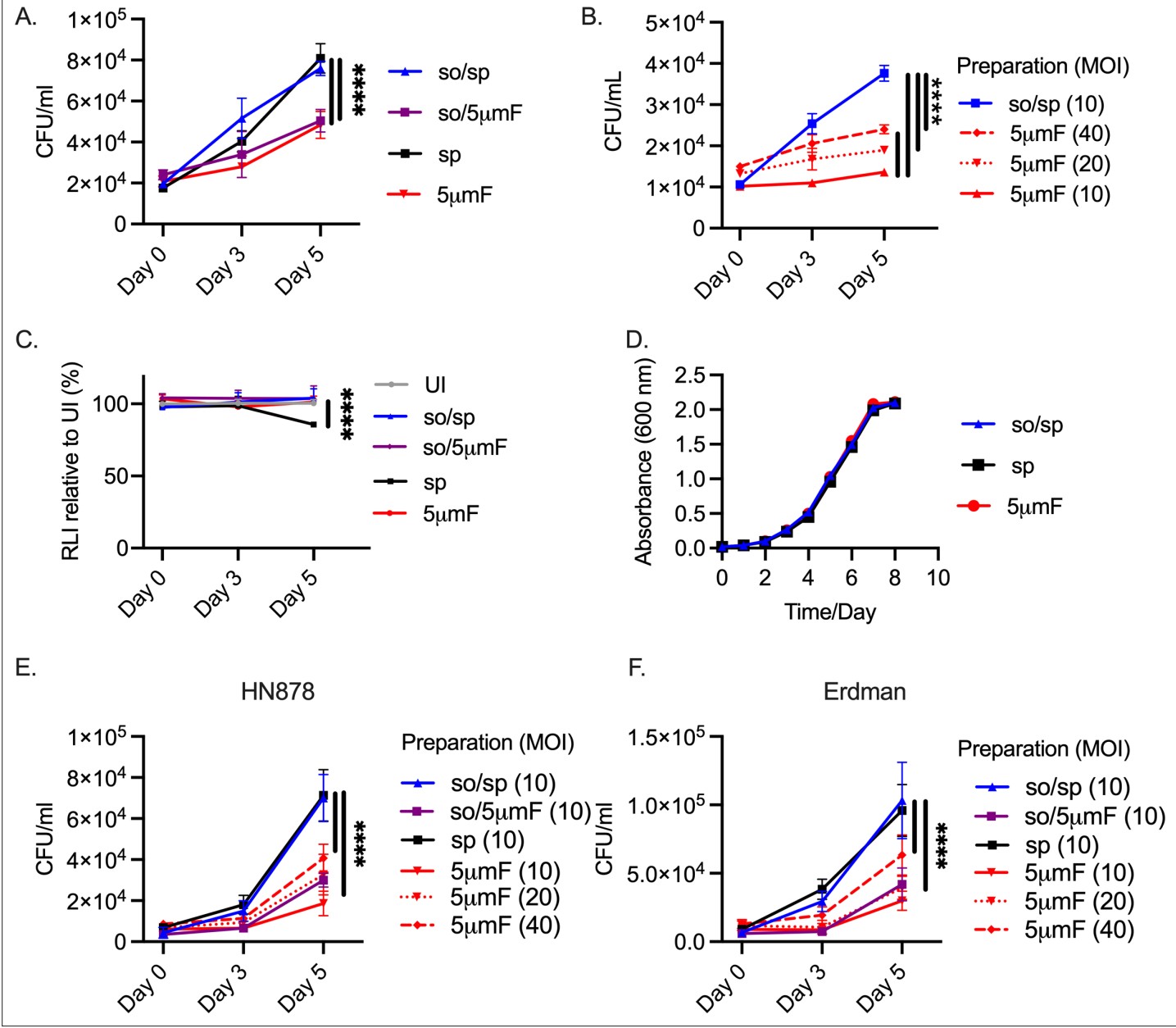

**Figure 4.** Filtered Mtb are attenuated in BMDMs. (**A**) BMDMs were infected with different preparations of Mtb (H37Rv) at an MOI of 10 and intracellular bacteria were enumerated by colony forming units (CFU) 4 hpi, 3 dpi, or 5 dpi. (**B**) BMDMs were infected with different preparations of Mtb (H37Rv) at an MOI of 10–40 and intracellular bacteria were enumerated by CFU 4 hpi, 3 dpi, or 5 dpi. (**C**) BMDMs were infected with different preparations of Mtb (H37Rv) at MOI of 10, and BMDM viability was measured using the CellTiter-Glo assay at 4 hpi, 3 dpi, and 5 dpi. Statistical significance between preparations was determined with two-way ANOVA using Dunnett's multiple comparisons test with selected significance values presented for 5 dpi relative to UI BMDMs. (**D**) Growth curve of different bacterial H37Rv preparation in liquid media (7H9 media supplemented with 10% Middlebrook OADC, 0.05% Tyloxapol, and 0.2% glycerol). (**E–F**) BMDMs were infected with different preparations of HN878 (**E**) or Erdman (**F**) strains at an MOI of 10–40 and intracellular bacteria were enumerated by CFU 4 hpi, 3 dpi, or 5 dpi. (**A–C, E–F**) For all CFU and macrophage viability studies, six biological replicates were used per group. Statistical significance was determined for each preparation at 5 dpi by comparing to CFU from BMDMs infected with spin-prepared Mtb using two-way ANOVA with Dunnett's multiple comparisons test. Selected significance values are presented at 5 dpi. (**A–F**) Error bars indicate mean +/-SD. ns not significant; ****<0.0001.

halo was restored by complementation, suggesting that lack of PDIM altered the interaction of uranyl acetate with the mycobacterial surface (*Figure 6D–I*). This difference, however, is unlikely to account for the hyperinflammatory signaling, as it was seen in all ΔppsD samples, and only the sonicated samples were hyperinflammatory. As we had seen with WT Mtb, there were round protrusions and

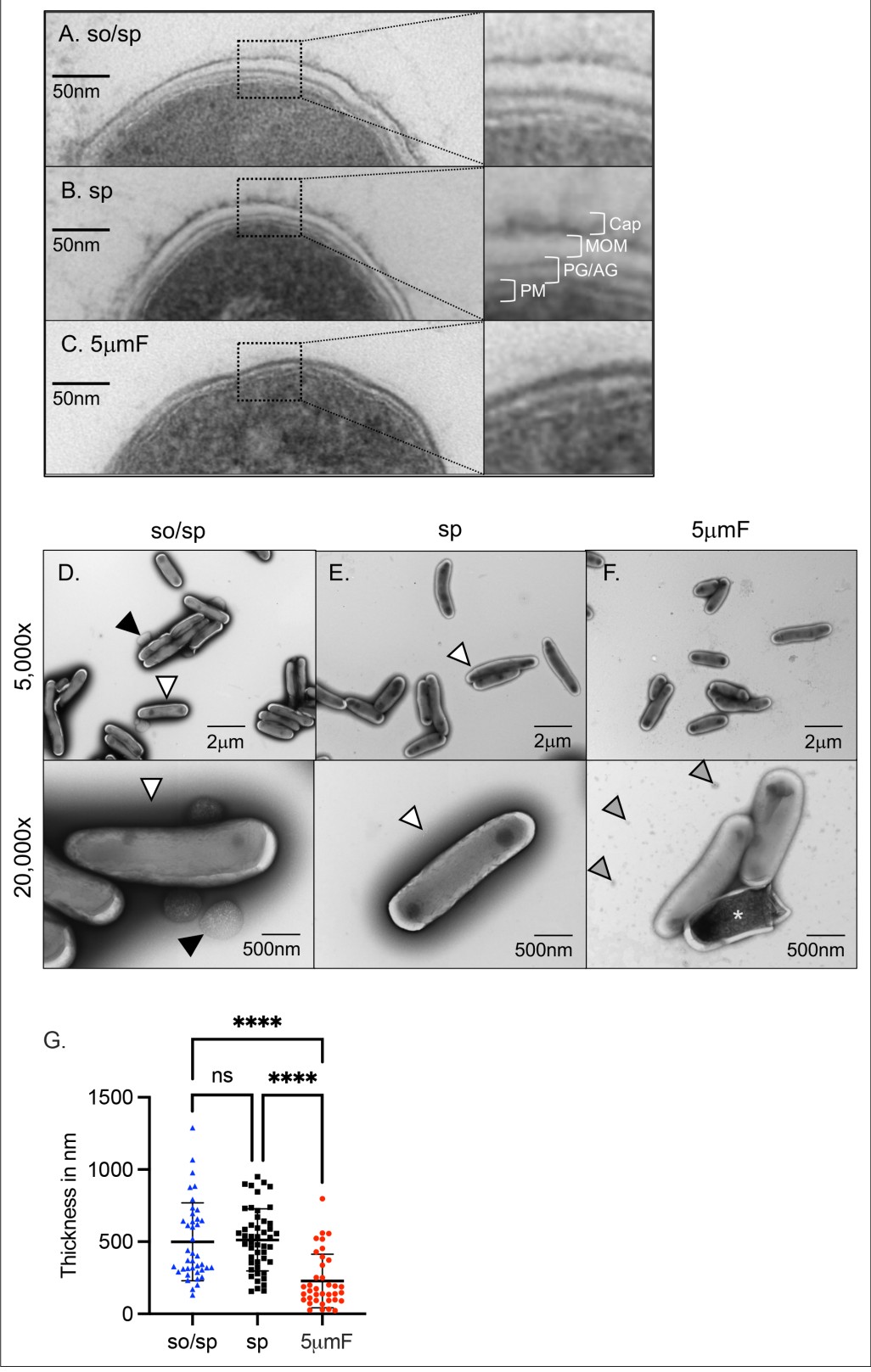

**Figure 5.** Sonication and filtering affect the bacterial cell wall. (**A–C**) TEM of ultrathin cross-sections of Mtb at ×50,000 magnification (left) beside enlarged cross-section of the envelope (right). The plasma membrane (PM), peptidoglycan/arabinogalactan layer (PG/AM), mycobacterial outer membrane (MOM), and capsular layer (Cap) are indicated. (**D–F**) Mtb were absorbed on freshly glow discharged formvar/carbon-coated copper grids followed

*Figure 5 continued on next page*

*Figure 5 continued*

by negative staining with 1% aqueous uranyl acetate. Representative images are 5000 x (above) and 20,000 x (below). So/sp-prepared Mtb had round protuberances that were on or near their envelopes indicated by black arrows. Electron-dense outer halos seen surrounding so/sp- and sp-prepared bacteria are indicated with white arrows. Debris seen in the extracellular space of 5µmF-prepared Mtb is indicated with gray arrows. (**G**) Capsule thickness was measured in nanometers using TEM images from bacteria stained with 1% uranyl acetate. Thickness measurements were compared between preparations with one-way ANOVA using Tukey's multiple comparisons test. Error bars indicate mean +/-SD. ns not significant; ****<0.0001.

vesicles in sonicated sample of both Δ*ppsD* and the complemented strain (*Figure 6D and G*). There was no obvious visual difference between the Δ*ppsD* mutant and complemented strain to explain why the so/sp Δ*ppsD* mutant was more hyperinflammatory than so/sp WT. To conclude, the hyperinflammatory phenotype associated with the Δ*ppsD* mutant depended upon the method of bacterial preparation.

## Discussion

More than 50 years ago, D'Arcy Hart demonstrated that *M. tuberculosis* avoids lysosomal delivery within macrophages (*Armstrong and Hart, 1971*). Ever since his landmark study, in an effort to understand fundamental mechanisms of TB pathogenesis, investigators have studied the interaction of Mtb with macrophages. They have also employed a variety of methods to disperse the bacilli to enable subsequent analysis. Unexpectedly, we found that two commonly used single cell preparation methods significantly impacted Mtb-host interactions: sonicated bacilli were hyperinflammatory, and 5µm-filtered Mtb were attenuated in macrophages. These effects were seen for H37Rv, HN878, and Erdman strains. In addition, we found that the method of preparation changes the impact of PDIM on the early macrophage transcriptional responses to Mtb. Consistent with the data of *Hinman et al., 2021*, who used a low-speed spin method to remove clumps, we found little impact of PDIM on the early TLR2-dependent response to centrifuged bacteria. However, if the bacteria were briefly sonicated then strains without PDIM elicited increased pro-inflammatory gene expression compared to control strains. This suggests that the mutant without PDIM is either more sensitive to sonication-induced damage or that it is more inflammatory once that damage occurs. It is important to point out that we only examined the early TLR2-dependent response. PDIM influences a variety of processes, including later TLR2-driven responses, phagosomal escape, and intracellular survival (*Augenstreich et al., 2017*; *Barczak et al., 2017*; *Cambier et al., 2020*; *Hinman et al., 2021*; *Lerner et al., 2018*; *Osman et al., 2020*; *Quigley et al., 2017*). It is possible that these other PDIM-dependent processes are not impacted by the preparation method. Nonetheless, our studies demonstrate that the preparation method needs to be considered in host-pathogen interaction studies, as it can change the interpretation of bacterial mutants and has a dramatic effect on TLR2-dependent responses and intracellular bacterial survival in bone marrow-derived macrophages.

Given the extensive use of macrophages in Mtb pathogenesis studies, there are surprisingly few studies investigating the impact of dispersal methods. A 2004 study demonstrated that prolonged sonication (5 min) reduces Mtb viability, while bacteria that had undergone gentle sonication (30 sec x 3) exhibited enhanced binding to macrophages and altered surface charge relative to syringed bacteria (*Stokes et al., 2004*). In that study, the sonicated bacteria had an altered cell envelope, which appeared uneven and bulging. Even though we sonicated for a shorter time (10 sec x 3), we also saw evidence of similar cell envelope disruption by TEM. In addition, we found that sonicated bacteria elicited orders-of-magnitude higher levels of TLR2-dependent transcriptional responses, leading to enhanced IL-1β and TNF-α secretion. This was mediated in part by material that was no longer cell associated, as even sterile-filtered samples activated macrophages. In addition, uninfected bystander cells that had been treated with so/sp preparations secreted TNF-α in the FluoroDOT assay. Our TEM findings suggest that sonication results in cell envelope damage and generation of small structures that resemble extracellular vesicles (EVs) that have been described in Mtb, although the vesicles that we saw are generally larger than the majority of EVs (*Prados-Rosales et al., 2011*). Mtb EVs are formed by an active process and contain immunomodulatory molecules including lipoarabinomannan and other TLR2 agonists (*Athman et al., 2015*; *Palacios et al., 2021*; *Prados-Rosales et al., 2011*).

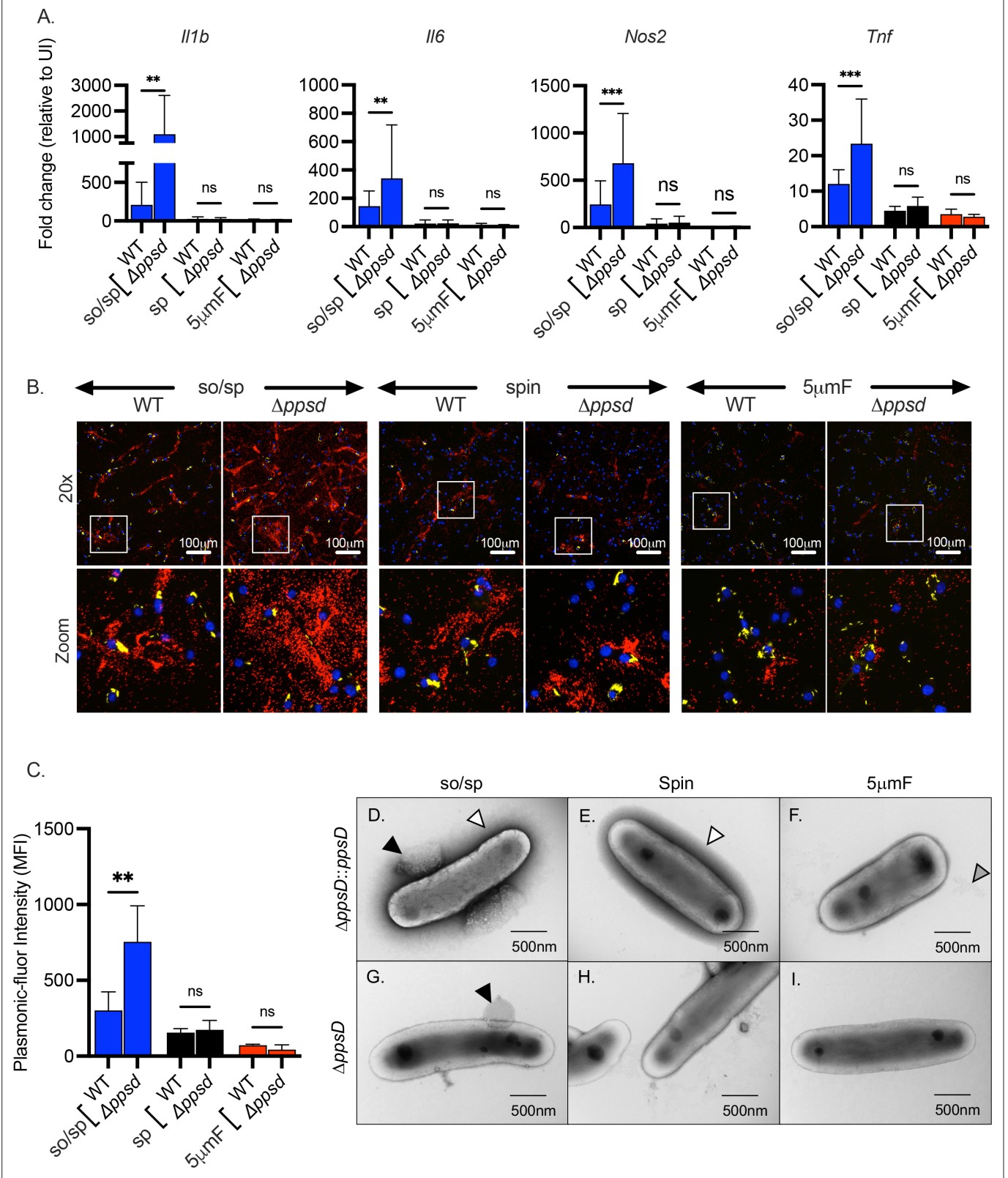

**Figure 6.** The role of PDIM in inflammatory responses depends upon preparation method. (**A**) BMDMs were uninfected or infected with indicated strains of Mtb at an MOI of 10, and gene expression was analyzed by qPCR at 6 hpi. Data are presented as fold change in gene expression relative to uninfected BMDMs of the same mouse genotype. Statistical significance was determined with two-way ANOVA using Tukey's multiple comparisons test. Data are combined from two to three experiments, each with three biological replicates per group and two technical replicates per sample. (**B**)

*Figure 6 continued on next page*

*Figure 6 continued*

Using the FluoroDOT assay, BMDMs were grown on a glass bottom plate that was coated with TNF-α capture antibody, infected at an MOI of 10 with H37Rv-GFP or ΔppsD-GFP prepared by the indicated method, and examined by epifluorescence microscopy (×20) 6 hpi. Images show Plasmonic-fluor 650 (red), Mtb (GFP), and DAPI (blue). Boxed areas in the image are enlarged in the bottom images. (**C**) Data show the quantification of the mean fluorescence intensity (MFI) of the plasmonic-fluor in the entire well from each conditions shown in B with statistical significance determined with two-way ANOVA using Tukey's multiple comparisons test. (**D**) Bacteria were imaged by allowing indicated Mtb strains to absorb on freshly glow discharged formvar/carbon-coated copper grids followed by negative staining with 1% aqueous uranyl acetate. Round protuberances seen on or near the envelopes of so/sp-prepared H37Rv Mtb are indicated by black arrows, the electron-dense outer halos seen surrounding so/sp- and sp-prepared H37Rv Mtb are indicated with white arrows, and the debris seen in 5μmF-prepared H37Rv Mtb are indicated with gray arrows. (**A, C**) Error bars indicate mean +/-SD. ns not significant; **p<0.01; ***<0.001.

Whether the structures formed by sonication have similar content to EVs found in growing cultures will require further studies.

How might sonication and filtration lead to the distinct macrophage responses that we observed? Electron microscopy revealed that sonication and filtration cause different types of alteration to the cell envelope (*Figure 7A*). The cell envelope is an elaborately layered structure that contains a variety of lipid and protein PAMPs and virulence factors. Our data are consistent with a model in which PAMPs and virulence factors are differentially impacted by sonication and filtration (*Figure 7B*). In this model, sonication disrupts the cell envelope in a manner that makes PAMPs more highly exposed or accessible, while leaving the activity of virulence factors intact. In contrast, filtration (5μmF) disrupts the cell envelope in such a way that both virulence factors and PAMPs are inactivated and/or dispersed from the bacilli, rendering the bacteria both attenuated and less inflammatory. Bacteria that are subject only to low-speed spin are neither hyper-inflammatory nor attenuated, since both PAMPs and virulence factors are less perturbed. In the case in which the bacteria are sonicated and filtered, they are both hyper-inflammatory and attenuated, which can be explained by enhanced exposure of PAMPs through sonication and inactivation of virulence factors by filtration.

This is one model that would explain our findings, but other scenarios could be envisioned. Mtb cultures are highly heterogenous, so we considered the possibility that a small minority of the so/sp bacilli were contributing to the heightened inflammatory response. However, the Fluoro-dot data, which allows us to visualize cytokine secretion at a single cell level, argue against this possibility. Similarly, if there were a small population of bacilli in the filtered sample that were inhibiting the inflammatory response, then, we would have expected the mixed samples to behave like the filtered sample, but rather we saw an intermediate phenotype. In terms of intracellular growth, there is undoubtedly heterogeneity within the population, with better intracellular growth on a population level in the so/sp and sp samples relative to filtration, with all samples having a mix of growth, stasis, and killing. Two aspects of heterogeneity that we assessed are clump size and capsule thickness. On a population level, we found a significant reduction in the thickness of the capsule of the 5μmF bacteria compared to both so/sp and sp bacteria. However, there was a wide distribution in cell envelope thickness, which may contribute to heterogeneity in macrophage responses to bacilli on a cell-to-cell level. Finally, we considered that differences in the degree of aggregation in the different bacterial preparations may account for differences in inflammatory potential or intracellular survival (*Rodel et al., 2021*), but heterogeneity in this aspect of the bacterial population was unlikely to explain the differences in outcomes. There may be other aspects of underlying heterogeneity in our samples that contribute to their distinct behavior or confound bulk measurements.

While our study was limited to three common single cell preparation methods, we expect that other techniques would also impact the mycobacterial envelope and host interactions. We queried PubMed for papers published in 2021 on Mtb and macrophages to determine which methods were commonly used (*Supplementary file 3*). Of the 119 papers, only 39.5% reported how they generated single cell suspensions. Of those that did report their methodology, 42.5% used more than one method. The most commonly reported methods were syringing, followed by sonication and low-speed centrifugation. Less often, filtering, vortexing with glass beads, or allowing gravity to sediment the larger clumps were used. Dispersing clumps with glass beads would likely disrupt the envelope, as studies have used this technique to selectively remove and isolate the capsular layer to analyze its components (*Lemassu and Daffé, 1994*; *Lemassu et al., 1996*; *Ortalo-Magné et al., 1995*). Others have reported that syringing through a 25-gauge needle produced no apparent disruption to the

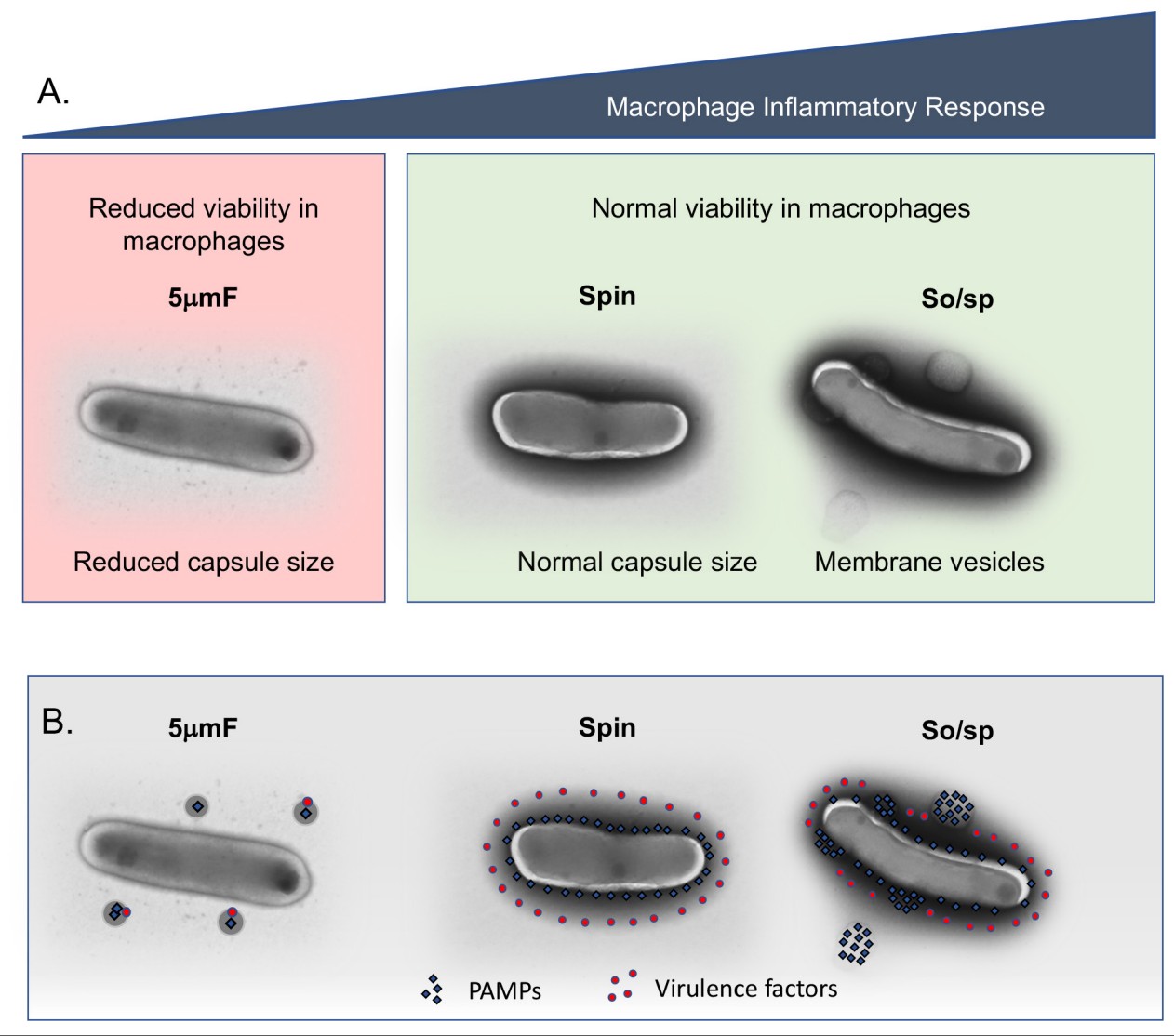

**Figure 7.** Summary of findings and model describing the impact of bacterial preparation methods on host-pathogen interactions. (**A**) Filtered (5μmF), spun , and sonicated (So/sp) Mtb differ in appearance and elicit different macrophage responses. 5μmF bacteria exhibit reduced growth in macrophages and have reduced capsular staining. Bacilli prepared with sonication elicit the strongest inflammatory response and have membrane vesicles and cell envelope protrusions. (**B**) The findings in (**A**) can be explained by the following model: filtration disrupts the cell envelope, dispersing and inactivating PAMPs and virulence factors, resulting in reduced macrophage inflammatory responses and reduced intracellular growth. Bacteria that are spun have an intact cell envelope that shields PAMPs and contains virulence factors. Sonication disrupts the cell envelope such that PAMPs are more highly exposed, resulting in increased inflammatory responses; virulence factors remain intact enabling normal growth in macrophages.

envelope on TEM (*Stokes et al., 2004*), but these samples were not evaluated further in terms of macrophage responses. We did not evaluate syringing, because it is not an approved method in our biosafety level 3 facility due to the risk of aerosolization and needle stick injuries. The physical forces used to disrupt clumps by this method might also result in envelope alterations, and investigators should consider this in their studies. Overall, we consider centrifuged samples as the least disrupted, but even centrifugation might disrupt the capsule, as do detergents that are commonly used in liquid cultures (and were used for all of our studies). Detergents are known to cause release of capsular components into the culture filtrate (*Kalscheuer et al., 2019*; *Sani et al., 2010*), although the impact on host interactions is relatively unexplored. The impact of detergent treatment on cytokine responses and vaccine responses have been evaluated, but after detergent treatment, single cell suspensions were generated by filtering, sonicating, or syringing (*Prados-Rosales et al., 2016*; *Sani et al., 2010*), complicating the interpretation.

Even if investigators had a non-disruptive way to isolate single cells, the behavior of single cells may not be the same as large aggregates that make up a substantial fraction of the unperturbed bacterial population. The aggregation state of Mtb has long been reported to be important to pathogenesis. For example, the observation that Mtb forms serpentine cords in vivo dates back to the earliest descriptions of the bacteria. Aggregated Mtb are found at the periphery of human necrotic granulomas, in alveolar macrophages of Mtb-infected patients, and are exhaled by infected individuals (*Dinkele et al., 2021*; *Rodel et al., 2021*; *Ufimtseva et al., 2018*). The literature describing the impact of aggregation on host interactions is difficult to interpret in light of our findings, as many of these studies used agitation with glass beads, filtration, sonication, or some combination of these procedures to generate the dispersed samples (*Kolloli et al., 2021*; *Mahamed et al., 2017*; *Rodel et al., 2021*). Thus, bacterial aggregation is likely an important virulence property of Mtb, which investigators overlook in the effort to generate single cell suspensions; at the same time, in generating single cell suspensions, investigators introduce the potential for experimental artifact.

It is possible that technical differences in how other laboratories sonicate or filter bacteria could result in findings that are different from ours. We used log phase cultures of H37Rv, HN878, and Erdman strains that had been grown with gentle agitation, a fatty acid source (oleic acid), and 0.05% Tyloxapol to infect mouse macrophages, but other investigators use different strains, frozen stocks, omit oleic acid, use different detergents, or infect human macrophages, all of which could lead to differences from our findings. An important conclusion of our findings is that investigators should fully report the methods that they use to grow and process mycobacteria and consider the impact of the methodology on their findings.

While sonication is an artificial stimulus, our findings suggest that Mtb keeps in check its massive pro-inflammatory potential by the organization and integrity of the envelope (*Figure 7B*). We imagine that by altering cell envelop architecture, Mtb tune their interactions to achieve the desired host response *Garcia-Vilanova et al., 2019*; for example, for initial infection and persistence, it may benefit the bacilli to minimize the TLR2-driven inflammatory response to promote immune evasion, whereas in order to drive tissue pathology and transmission, the bacilli may generate a hyperinflammatory phenotype (*Chandra et al., 2022*). While the Mtb cell wall is known to be dynamic (*Dulberger et al., 2020*), little is known about the structure and function of the cell wall during different in vivo contexts. To this end, a recent study evaluated the ultrastructure of the Mtb cell wall ex vivo from infected human sputum samples (*Vijay et al., 2017*). The characteristic three layers were found, and a reduction in the electron translucent layer was noted when bacilli were grown under stress conditions. Further in vivo studies investigating how Mtb regulates cell envelope architecture to modulate host inflammatory responses and deploy virulence lipids and protein effectors are needed.

## Materials and methods
### Bacterial strains and growth conditions

The Mtb strains H37Rv (WT), Δ*ppsD*, and Δ*ppsD::ppsD* were used in this study. The Δ*ppsD* and Δ*ppsD::ppsD* strains were from A. Barczak and previously described (*Barczak et al., 2017*). HN878 strain was a gift from Shabaana Khader, and Erdman was from Christina Stallings. Bacteria were grown to mid-log phase in an incubator at 37 °C with 5% $CO_2$ and gentle agitation (120 rpm). Bacteria were grown in 7H9 media supplemented with Middlebrook OADC (oleic acid, albumin, dextrose, catalase), 0.05% Tyloxapol, and 0.2% glycerol. H37Rv Δ*ppsD* growth media was additionally supplemented with 50 µg/mL hygromycin, GFP-expressing bacterial strains with 25 µg/mL kanamycin, and Δ*ppsD::ppsD* with 50 µg/mL hygromycin and 25 µg/mL kanamycin.

### Generation of single cell suspensions of Mtb

Following growth of Mtb to mid-log phase ($OD_{600}$ 0.5–0.8), bacteria were washed with phosphate-buffered saline (PBS) and resuspended in the appropriate media for the subsequent study. Single cell suspensions of Mtb were generated using one or a combination of the following methods: (1) low-speed spin (sp): bacteria were centrifuged at 206 x g for 10 min followed by 132 x g for 8 min, with the supernatant collected following each spin; (2) 5 µm filter (5µmF): 6–20 ml of bacterial culture were added to a 10 mL syringe and then, with gentle pressure applied to the syringe plunger, passed through a 5 µm polyethersulfone (PES) filter (PALL Life Sciences; cat. 4650) except

in the case were polytetrafluoroethylene (PTFE) filters (Tisch scientific; cat. SF17400) were used; (3) sonication (so): 4–10 ml bacteria in a 15 mL conical tube were placed in a water bath sonicator (Branson Ultrasonics Corporation, Digital Sonifier 450) and sonicated with three pulses lasting 10 s each, with an amplitude of 70% and 5 s rests between each pulse. Following sonication, bacteria were centrifuged with a low-speed spin (so/sp) or passed through a 5 µm filter (so/5µmF), as described above. Following the preparations described above, the concentrations of bacterial suspensions with $OD_{600}$ between 0.04 and 0.12 were calculated using the formula: 1 $OD_{600}$=3 x $10^8$ bacteria per mL (except filtered bacterial suspensions, which were calculated using: 1 $OD_{600}$=2 x $10^8$ bacteria per mL). The multiplicity of infection was determined by plating the input on 7H10 plates. For some experiments, bacterial cultures were further passed through a 0.2 µm filter (PALL Life Sciences; cat. 4652).

## Mice

Eight- to 12-week-old male and female C57BL/6 J and $Tlr2^{-/-}$ (B6.129-Tlr2tm1Kir/J; Strain #004650) mice were obtained from The Jackson Laboratory. All work with mice were approved by the Washington University School of Medicine Institutional Animal Care and Use Committee (protocol # 21–0245). Euthanasia was performed prior to bone marrow harvest in accordance with the 2020 *AVMA Guidelines for the Euthanasia of Animals* prior to tissue harvest.

## Bone marrow-derived macrophage isolation and infection

Mouse hematopoietic stem cells were isolated as described in *Banaiee et al., 2006*. Hematopoietic cells were differentiated by culturing for 7 days in Dulbecco's Modified Eagle Medium (DMEM) with 10% FBS, 2 mM L-glutamine, and 1 mM pyruvate (DMEM complete). DMEM complete media was supplemented with 20% L929 cell supernatant (as a source of macrophage colony stimulating factor (M-CSF), 100 U/mL final concentration), 10 units/ml penicillin, and 10 units/ml streptomycin. Following differentiation, BMDMs were washed with PBS, resuspended in DMEM complete with 10% L929 cell supernatant, and plated for infection the following day. Single cell suspensions of Mtb in DMEM complete with 10% L929 added to macrophages at a MOI of 5, 10, 20, or 40, and plates were spun for 5 min at 51 x g. The MOI was verified by plating the inoculum. At 4 hpi, macrophages were washed 3 times with DMEM to remove extracellular bacteria. To enumerate CFU, at specified time points macrophages were lysed with 0.06% sodium dodecyl sulfate (SDS) in water and serially diluted in PBS. The cell lysates were plated on 7H11 agar plates supplemented with OADC and glycerol, and CFU were counted after 14–21 days. For qPCR, macrophages were lysed in TRI Reagent (Zymo Research, R2050-1-50), and total RNA was extracted.

## RNA sequencing

Mouse hematopoietic cells were collected and differentiated to BMDMs as above. $1.6x10^6$ BMDMs per well were incubated overnight in a six-well plate. BMDMs were either uninfected or infected at a MOI of 5 with Mtb prepared by the designated single cell preparation method. Five samples per group were used. At 72 hpi, macrophages were lysed in TRI Reagent and total RNA was extracted. Total RNA integrity was determined using Agilent Bioanalyzer or 4200 Tapestation. Library preparation was performed with 500 ng to 1 µg total RNA. Ribosomal RNA was removed by an Rnase-H method using RiboErase kits (Kapa Biosystems). mRNA was then fragmented in reverse transcriptase buffer and heating to 94 °C for 8 min. mRNA was reverse transcribed to yield cDNA using SuperScript III RT enzyme (Life Technologies, per manufacturer's instructions) and random hexamers. A second strand reaction was performed to yield ds-cDNA. cDNA was blunt ended, had an A base added to the 3' ends, and then had Illumina sequencing adapters ligated to the ends. Ligated fragments were then amplified for 12–15 cycles using primers incorporating unique dual index tags. Fragments were sequenced on an Illumina NovaSeq-6000 using paired end reads extending 150 bases. The raw CPM values that were generated underwent filtering, with removal of mitochondrial RNA, autosomal rRNA, and low-expressed genes with less than 1 CPM in the smallest group size, followed by Voom transformation of counts. Differentially expressed genes were then determined using the 'limma' package from bioconductor.org. Heatmaps were generated in R using the 'pheatmap' package.

## Gene set enrichment analysis

We imputed normalized gene expression data and associated Ensembl Stable IDs of differentially expressed genes from our RNA-seq experiment into GSEA software. GSEA then analyzed our dataset for enriched genetic signatures curated in the hallmark gene sets by the Molecular Signatures Database (*Liberzon et al., 2015*; *Subramanian et al., 2005*). Genes were ranked based on their expression and compared against the hallmark gene sets in order to generate an enrichment score. A nominal p value was then generated followed by normalization for the size of the gene set and adjustment for multiple hypothesis testing to yield a false discovery rate (FDR) (*Subramanian et al., 2005*). Gene sets which had a p-value <0.01 and an FDR <0.01 were considered significant.

## Quantitative polymerase chain reaction

BMDMs ($2.0 \times 10^5$ per well) were plated in 24-well plates. At indicated time points, macrophage growth media was aspirated, and 100 μL TRIzol (Zymo Research, R2050-1-50) was added to each well followed by isolation of total RNA using Direct-Zol RNA Mini-Prep Plus Kit (Zymo Research, R1058) according to the manufacturer's instructions. RNA concentrations were determined using NanoDrop One (Thermo Fisher Scientific Inc, Waltham, MA), and cDNA was made with High Capacitance cDNA Reverse Transcription Kit (Thermo Fisher Scientific Inc, Waltham, MA). Quantitative PCR was performed using SYBR Green dye (CFX Connect Real-Time System, Bio-Rad Laboratories, Inc, Hercules, CA). Fold changes in gene expression were calculating by normalizing data to *Gapdh* as a house-keeping gene and values were presented relative to uninfected cells. The nucleotide sequences of all primers used are presented in *Supplementary file 2*.

## Macrophage viability assay

BMDMs were plated in 200 μL of media in a 96-well white optical plate. After BMDMs were allowed to adhere, cells were infected with Mtb of the appropriate preparation at an MOI of 10. The plates were centrifuged at 51 x g for 5 min and incubated at 37 °C and 5% $CO_2$. At 4 hpi, macrophages were washed three times with DMEM to remove extracellular bacteria. At the appropriate time points, macrophage viability was determined using the CellTiter-Glo Luminescent Cell Viability Assay (Promega, catalog number G7570). At 4 hpi, and 3 and 5 dpi, the media was aspirated and a solution of 100 μL DMEM and 25 μL CellTiter-Glo solution was added to each well per the manufacturer's instructions. The plates were incubated at 37 °C for 10 min, covered with optical tape, and luminescence was then determined using a Synergy HTX Multi-Mode Reader (Agilent Technologies, Inc, Santa Clara, CA). The relative luminescence units were normalized to the reading for uninfected BMDM samples at each given timepoint. Six biological replicates per group were performed at each timepoint.

## Fluorescence microscopy

BMDMs ($3 \times 10^4$ per well) were seeded in glass bottom 96-well plate (Ibidi, catalog number 89626) and infected with GFP-expressing H37Rv at a MOI of 5. After 4 hr, macrophages were washed with PBS and fixed with 1% paraformaldehyde in PBS overnight followed by permeabilization in 0.1% vol/vol Triton X-100 (Millipore Sigma) in PBS for 10 min at room temperature (RT) and blocked for 45 min in 2% bovine serum albumin (BSA) in PBS prior to staining with DAPI (4=,6-diamidino-2-phenylindole) and mounted in Prolong Diamond antifade (Molecular Probes, Life Technologies). Images were captured using a Nikon Eclipse Ti confocal microscope (Nikon Instruments, Inc, Melville, NY) equipped with a 60 X apochromat oil objective lens. Image acquisition was done using NIS-Elements version 4.40. Fluorescent images were then used to further analyze the bacterial clumps based upon manual quantification of GFP-bacteria in infected macrophages. Single bacteria and clumps of bacteria were quantified for each of the preparations and at least 100 bacterial occurrences were analyzed for each preparation method.

## ELISA

$2.0 \times 10^5$ BMDM in 24-well plates were infected with Mtb at a MOI of 10. At the indicated timepoints, the cell supernatant was collected and filtered through 0.22 μm filters. Cytokines were measured from the supernatant with R&D Systems DuoSet ELISA kits for TNF-α and IL-1ß, according to the manufacturer's instructions (R&D Systems, cat. DY410, DY401). Three biological replicates per group and

two technical replicates per sample were used, and experiments were repeated at least two times per experimental condition.

## FluoroDOT assay

Assays were performed using reagents from Mouse TNF-α DuoSet ELISA kits (R&D systems, catalog number DY410-05). Wells of 96-well glass-bottom, black plates (P96-1.5H-N, Cellvis, Mountain View, USA) were coated with 100 µL TNF-α capture antibody (2 µg/mL in PBS) at 4 °C overnight. Coated wells were then washed three times with PBS, followed by blocking with reagent diluent (0.2 µm filtered 1% BSA in PBS) for at least 1 hr at RT. Wells were washed three times with PBS and thereafter $8.0 \times 10^3$ BMDMs in DMEM complete with 10% L cell supernatant were added to each well. The same day, BMDMs were infected with the indicated GFP-expressing Mtb strains that had prepared as single cell suspensions. Macrophages were incubated at 37 °C in 5% $CO_2$ for 3 hr, followed by three washes with fresh media to remove extracellular Mtb, and incubated in the same media for an additional 3 hr. Media was then aspirated and 200 µL 4% PFA in PBS was added for 30 min at 37 °C. Wells were washed with PBS and incubated with biotinylated 75 ng/mL TNF-α detection antibody in reagent diluent for 2 hr at RT. Wells were washed three times with PBS followed by 100 µL PBS containing streptavidin plasmonic-fluor 650 (PF650, extinction 0.5; Auragent Bioscience LLC; *Wang et al., 2021*) for 30 min at RT in the dark. Cells were washed three times with PBS and stained with 300 nM DAPI (Millipore Sigma) for 5 min at RT in the dark. Wells were washed three times with PBS and then visualized using a Nikon TsR2 epifluorescence microscope.

## Transmission electron microscopy

Bacteria were grown to mid-log phase, and single cell suspensions were generated in PBS as described above. Bacteria were incubated in 4% PFA for 30 min at 37 °C followed by centrifugation at 3000 x *g* and resuspension in PBS. For ultrastructural analyses using ultrathin cross-sections through bacteria, samples were further fixed in 2% paraformaldehyde/2.5% glutaraldehyde (Ted Pella Inc, Redding, CA) in 100 mM sodium cacodylate buffer, pH 7.2 for 2 h at RT and then overnight at 4 °C. Samples were washed in sodium cacodylate buffer at RT and postfixed in 2% osmium tetroxide (Ted Pella Inc) for 1 hr at RT. Samples were then rinsed in dH20, dehydrated in a graded series of ethanol, and embedded in Eponate 12 resin (Ted Pella Inc). Sections of 95 nm were cut with a Leica Ultracut UCT ultramicrotome (Leica Microsystems Inc, Bannockburn, IL), stained with uranyl acetate and lead citrate, and viewed on a JEOL 1200 EX transmission electron microscope (JEOL USA Inc, Peabody, MA) equipped with an AMT 8-megapixel digital camera and AMT Image Capture Engine V602 software (Advanced Microscopy Techniques, Woburn, MA).

For imaging of whole bacteria, after bacterial samples were fixed with 4% PFA, they were allowed to adsorb onto freshly glow discharged formvar/carbon-coated copper grids for 10 min. Grids were then washed in dH2O and stained with 1% aqueous uranyl acetate (Ted Pella Inc, Redding, CA) for 1 min. Excess liquid was gently wicked off, and grids were allowed to air dry. Samples were viewed by transmission electron microscopy as described above. TEM images were then used to further analyze the bacteria using ImageJ (version 1.53q). Single bacteria and clumps of bacteria which were completely contained within the image borders were further analyzed for each of the preparations. For each single bacteria or clump, a measurement was then taken of the thickness of the capsular layer. Lines were drawn perpendicular to the middle of the long axis of the bacteria, capturing the black-staining outer component.

## Methodology of literature review

We conducted a search in PubMed using the medical subject headings '*Mycobacterium tuberculosis*' and 'Macrophage' and filtered for articles published in 2021. This generated 183 articles, which were further filtered to include only original research articles by removing review articles, protocols, and commentaries. The remaining 155 articles were included in the analysis if they performed an in vitro macrophage infection with live Mtb. The text, supplementary methods, and figures were reviewed to determine the single cell preparation methods used.

## Statistical analysis

Graph Pad Prism 9 software was used for statistical analysis and to prepare graphs. Error bars used in the figures correspond to the mean and standard deviation. Statistical significance was determined using unpaired T test, one-way analysis of variance (ANOVA), or two-way ANOVA, as indicated.

## Acknowledgements

We thank members of the Philips laboratory for their input. Funding for these studies came from NIAID/NIH (R01 AI087682 and AI30454) to JAP, and the National Cancer Institute (NCI)-Innovative Molecular Analysis Technologies (R21CA236652) and National Science Foundation (CBET-1900277) to SS. ATR was supported by NIH/NHLBI (T32 HL007317-37). We thank the Genome Technology Access Center at Washington University School of Medicine for help with genomic analysis. The Center is partially supported by NCI Cancer Center Support Grant #P30 CA91842 to the Siteman Cancer Center and by ICTS/CTSA Grant# UL1TR002345 from the National Center for Research Resources (NCRR), a component of the National Institutes of Health (NIH), and NIH Roadmap for Medical Research. This publication is solely the responsibility of the authors and does not necessarily represent the official view of NCRR or NIH.

## Additional information

### Competing interests

Anushree Seth: is currently employed with Auragent Bioscience LLC. The plasmonic-fluor technology used in the manuscript has been licensed by the Office of Technology Management at Washington University in St. Louis to Auragent Bioscience LLC. Srikanth Singamaneni: is an inventor on a provisional patent related to plasmonic-fluor technology, and the technology has been licensed by the Office of Technology Management at Washington University in St. Louis to Auragent Bioscience LLC. SS is a co-founder/shareholder of Auragent Bioscience LLC. SS along with Washington University may have financial gain through Auragent Bioscience LLC through this licensing agreement. These potential conflicts of interest have been disclosed and are being managed by Washington University in St. Louis. The other authors declare that no competing interests exist.

### Funding

| Funder | Grant reference number | Author |
|---|---|---|
| NIAID/NIH | R01 AI087682 | Jennifer A Philips |
| NIAID/NIH | R01 AI30454 | Jennifer A Philips |
| National Cancer Institute (NCI)-Innovative Molecular Analysis Technologies | R21CA236652 | Srikanth Singamaneni |
| National Science Foundation | CBET-1900277 | Srikanth Singamaneni |
| NIH/NHLBI | T32 HL007317-37 | Andrew T Roth |

The funders had no role in study design, data collection and interpretation, or the decision to submit the work for publication.

### Author contributions

Ekansh Mittal, Conceptualization, Formal analysis, Investigation, Visualization, Methodology, Writing - original draft, Writing – review and editing; Andrew T Roth, Formal analysis, Investigation, Visualization, Writing - original draft, Writing – review and editing; Anushree Seth, Investigation, Visualization, Methodology, Writing – review and editing; Srikanth Singamaneni, Resources, Supervision, Funding acquisition, Methodology, Writing – review and editing; Wandy Beatty, Resources, Formal analysis, Visualization, Methodology, Writing – review and editing; Jennifer A Philips, Conceptualization,

Resources, Formal analysis, Supervision, Funding acquisition, Visualization, Writing - original draft, Project administration, Writing – review and editing

### Author ORCIDs
Ekansh Mittal  http://orcid.org/0000-0001-9034-033X
Andrew T Roth  http://orcid.org/0000-0003-4239-7926
Jennifer A Philips  http://orcid.org/0000-0002-9476-0240

### Ethics
All work with mice were approved by the Washington University School of Medicine Institutional Animal Care and Use Committee (IACUC protocol # 21-0245). Euthanasia was performed prior to bone marrow harvest in accordance with the 2020 AVMA Guidelines for the Euthanasia of Animals prior to tissue harvest.

### Decision letter and Author response
Decision letter https://doi.org/10.7554/eLife.85416.sa1
Author response https://doi.org/10.7554/eLife.85416.sa2

## Additional files

### Supplementary files
• Supplementary file 1. Table of macrophage genes differentially expressed based upon preparation method. This file lists the genes that were differentially expressed between uninfected macrophages, macrophages infected with Mtb prepared by sonication followed by low-speed spin (so/sp), and macrophages infected with Mtb prepared by passing through a 5 μm filter (5μmF). Infectious were carried out at an MOI of 5 and analyzed at 72 hpi.

• Supplementary file 2. PCR primers used.

• Supplementary file 3. Literature review of methods used to generate single cell Mtb suspensions. (A) Approach used to analyze the literature to define the frequency with which distinct single cell preparation methods are used and how often they are reported. (B) Graph demonstrates the distribution of methods reported. Since some studies used multiple methods, the total does not equal 100.

• MDAR checklist

### Data availability
RNA-seq data can be accessed in BioProject (Accession PRJNA851060; ID: 851060).

The following dataset was generated:

| Author(s) | Year | Dataset title | Dataset URL | Database and Identifier |
| --- | --- | --- | --- | --- |
| Philips JA, Mittal E, Roth AT | 2022 | Single cell preparations of Mycobacterium tuberculosis damage the mycobacterial envelope and disrupt macrophage interactions (house mouse) | https://www.ncbi.nlm.nih.gov/bioproject/PRJNA851060 | NCBI BioProject, PRJNA851060 |

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
