## [Editor Report]

This important study examines how the method used to prepare *Mycobacterium tuberculosis* bacilli for use in experimental infection models affects outcomes. The authors provide compelling evidence indicating that the method of preparation affects the bacterial cell wall and substantially influences both host responses and infection outcomes in models of TB infection. The data will be useful for interpreting published TB literature and for future studies probing *M. tuberculosis* virulence. The work will be of interest to tuberculosis researchers and microbiologists in general.

---

## [Decision Letter]

**Decision letter after peer review:**

[Editors’ note: the authors submitted for reconsideration following the decision after peer review. What follows is the decision letter after the first round of review.]

Thank you for submitting the paper "Single cell preparations of *Mycobacterium tuberculosis* damage the mycobacterial envelope and disrupt macrophage interactions" for consideration by *eLife*. Your article has been reviewed by 3 peer reviewers, and the evaluation has been overseen by a Reviewing Editor and a Senior Editor. The reviewers have opted to remain anonymous.

Comments to the Authors:

We are sorry to say that, after consultation with the reviewers, we have decided that this work will not be considered further for publication by *eLife*.

Specifically, reviewers felt that your submission reports potentially important data however, the observations require further mechanistic insight to serve as a useful resource for the field. More detail is provided in the reviewer comments below.

*Reviewer #1 (Recommendations for the authors):*

The mycobacterial cell wall is complex, including an outer mycomembrane, non-covalently attached complex lipids, and depending upon growth conditions, a polysaccharide capsule. Multiple constituents of the cell wall have been shown to have impacts on host cells. In this manuscript, the authors investigate whether different methods used to prepare a single cell suspension of Mtb for infection studies change the mycobacterial cell wall in ways that lead to differential outcomes after infection. To do so, they compared sonication of bacteria followed by a low-speed spin with filtration through a 5uM filter. Sonicating followed by a low-speed spin resulted in an infection that was more inflammatory with higher induction of IL1, IL6, Nos2, and TNF. This increased inflammatory cytokine expression was TLR2 dependent. In addition, they show that sonicated bacteria grow in macrophages whereas filtered bacteria do not. However, the method of preparation of bacteria has no effect in on growth or inflammatory cytokine production in vivo in the mouse model. Finally, they show that mutants lacking the complex lipid PDIM are hyperinflammatory relative to wild type, as has been previously reported, but only when bacteria are prepared by sonication prior to infection. The major strength of this paper is the potential for providing clarity for the entire field around the impact of technical methods for preparing bacteria for infection during macrophage studies, which could bring more consistency and ability to interpret disparate results. The weaknesses are a lack of biological mechanistic insight, the lack of impact on in vivo infections and inconsistencies that make it difficult to draw clear conclusions from the data presented.

1. It is interesting that the authors present data showing a difference in growth in bone marrow derived macrophages in which sonicated or spun bacteria can grow, but filtered bacteria do no. Many labs do not sonicate bacterial stocks prior to macrophage infection, and even filter the cells, yet do see growth of bacteria in macrophages. In fact, from at least one previous publication from the author's lab, although the methods report that 5uM filtering was used to prepare the bacteria for macrophage infection, they observed growth of the bacteria in macrophages (PMID: 32636249). Can the authors explain this apparent discrepancy?

2. There are differences in the way bacteria that have been spun only (sp) behave compared with sonicated and spun (so/sp) vs filtered that make a clear understanding of the impact of these treatments on the bacteria difficult. For example, sp bacteria most closely resemble filtered when comparing inflammatory cytokine gene expression; however like so/sp bacteria they grow in macrophages. Sp bacteria also retain an outer layer (the composition of which is ambiguous) that so/sp bacteria do but do not have membrane blebs. Thus taken together, it is difficult to connect the observed differences in bacterial morphology with the different phenotypes seen during macrophage infection.

3. The authors show data that PDIM mutants are hyper-inflammatory relative to wild-type bacteria, but only if they have been sonicated. This is contradictory to a previous publication that showed that PDIM mutants elicit higher levels of TNFa (also examined in this study) during macrophage infections in which bacteria were not prepared by sonication (PMID: 34755600).

4. The authors present data showing that there is pro-inflammatory activity contained within the supernatants of bacteria that have been sonicated by applying supernatants from a bacterial suspension that has been passed through a 0.2uM filter to macrophages. However, it is not clear how much of the observed pro-inflammatory activity during infection itself comes from these soluble materials vs the bacteria themselves. If they wash the sonicated bacteria what do they see? It would be nice to see an experiment with so/sp bacteria, sups, and washed so/sp bacteria side by side.

*Reviewer #2 (Recommendations for the authors):*

The interaction of Mtb with macrophages is extensively studied by TB investigators and several studies have suggested that the methodology used to prepare Mtb for macrophage infections may influence the macrophage response. In this study the authors aimed to carefully examine this phenomenon with the lab-derived Mtb H37Rv strain. They found that disaggregation methods had a huge impact on macrophage responses in vitro and cautioned that this may lead to experimental artefacts. However, Mtb subjected to different disaggregation methods showed no difference in their in vivo growth and cytokine production in infected mice.

Strengths

1. The authors tackle an important problem that TB investigators face when performing in vitro macrophage infection studies that may lead to erroneous interpretation of data.

2. The data support the conclusions drawn.

3. TEM studies to examine damage to bacterial envelop by the two disintegration methods.

Weaknesses

1. The authors have not considered bacterial cell population heterogeneity.

2. Lack of a mechanism for the different in vitro and in vivo outcomes with sonicated and filtered Mtb.

3. No data with Mtb clinical strains.

In this study, the authors examined the impact of two routinely used bacterial disaggregation techniques on the subsequent interaction of Mtb with macrophages. They found that sonicated bacilli induced a strong hyperinflammatory response in BMDMs and still replicated normally in the macrophages. In stark contrast, Mtb that had been passed through a filter induced little inflammatory response in BMDMs and were also highly growth attenuated. Absence of PDIM impacted sonicated but not filtered bacteria. Interestingly, disaggregation methods did not impact growth in AMs or in vivo. In conclusion, the data presented here indicate that macrophage responses are strongly affected by the bacterial disaggregation methodology used and this may confound the results of Mtb-macrophage interaction studies.

Comments

1. The authors should rule out the possibility that their findings could be biased by cell population heterogeneity in Mtb H37Rv.

2. A lipid analysis of the bacteria after sonication and filtration would help substantiate the authors' conclusions.

3. Data from additional Mtb strains is required to strengthen the observations.

*Reviewer #3 (Recommendations for the authors):*

In this work, Mittal et al. determine how three commonly used methods for preparing Mtb for use in cellular and animal models of infection impact host response and infection outcomes. While largely technical, the key findings of the manuscript are important for understanding the existing literature of TB infection biology, and will undoubtedly inform how groups studying Mtb virulence in cellular and animal models perform and interpret their experiments in the future.

The primary comparisons were between sonication, filtration through a 5um filter, and centrifugation. Key findings:

1) Macrophages responded markedly differently to Mtb that were sonicated, filtered to make single-cell suspensions, or centrifuged- the primary initial outcome evaluated was comprehensive transcriptional profiling at 72 hours.

2) Mtb preparation using sonication results in enhanced inflammatory responses relative to uninfected or filtered cells. Outcomes evaluated include focused inflammatory gene transcriptional responses at 6 hours and 24 hours and TNF secretion as measured by FluoroDOT assays and TNF secretion and IL-1b secretion as measured by ELISA. Further, this enhanced inflammation is conferred by the sterile filtrate of the sonicated Mtb preparation, suggesting shedding of bacterial components into the media.

3) Mtb preparation using filtration results in reduced growth in bone marrow-derived macrophages but no impaired growth in alveolar macrophages or in vivo in mice.

4) TEM and uranyl acetate staining showed structural differences in the outer layers of Mtb prepared using sonication, filtration, and centrifugation. Sonicated Mtb had associated extra-bacterial vesicles; filtered Mtb had a thin outer coat suggesting loss of the outer membrane and capsule.

5) The impact of PDIM on early macrophage inflammatory responses as measured by targeted inflammatory gene expression and TNF FluorDOT assay was notable only for sonicated Mtb.

The finding that the preparation of Mtb changes inflammatory responses is compelling and well-supported by the transcriptional and cytokine data, in particular for the enhanced response to sonicated bacteria. The findings that this enhanced inflammation is partially TLR2 dependent and is associated with a soluble component of the bacterial preparation are similarly solid.

The finding that filtration results in reduced Mtb growth in BMDM relative to sonicated or centrifuged Mtb would be stronger if macrophage survival was assayed in parallel. At the MOIs used, Mtb infection is well-described to result in some degree of macrophage cell death, which can significantly impact recovery of Mtb from macrophage cell cultures. Clumping has also been described to have implications for macrophage cell death; given that the authors show different levels of clumping for bacteria prepared using each method, this is a particularly important outcome to measure in parallel with CFU.

The interpretation that the TEM findings and uranyl acetate staining (small extra bacterial vesicles for sonicated Mtb and loss of membrane/capsule for filtered Mtb) likely reflect the underlying reasons for differences in macrophage responses to the bacteria is consistent with the data, but not explicitly tested (and not readily testable).

The interpretation that differences in pre-existing macrophage inflammatory state in BMDM vs. alveolar macrophages drive the differences observed in the intracellular growth of filtered Mtb is also consistent with the data but not explicitly tested.

The overall conclusion that the specific methods used to prepare Mtb for infection impact macrophage responses and infection outcomes is solid and supported by the data. The conclusion that the method of bacterial preparation should be considered in both interpreting existing literature and experimental planning is also well-founded based on the data shown.

1) No testing of macrophage viability: Macrophage viability is another critical virulence related infection outcome, and can confound CFU measurements- in particular at higher MOIs and later timepoints. Measuring macrophage viability in parallel with CFU measurements would provide an important control for the CFU data and would offer insight into how the method of Mtb preparation impacts this additional infection outcome metric.

2) No testing of bacterial viability: While the relatively similar day 0 CFU in the macrophage infection assays suggest that similar numbers of live bacteria come out of each preparation method, the data would be stronger if bacterial viability were explicitly tested immediately after each preparation (using a live-dead reporter strain, or even CFU of the in vitro bacteria).

3) The hypothesis that the baseline inflammatory state of the macrophage influences whether filtered Mtb can grow is testable and potentially worth testing to more definitively support this idea. This could be done by pre-stimulating the macrophages with a range of that change the inflammatory and/or polarization state of the cells, then infecting with Mtb prepared using the three methods profiled.

---

## [Author Response]

[Editors’ note: the authors resubmitted a revised version of the paper for consideration. What follows is the authors’ response to the first round of review.]

Essential revisions:Reviewer #1 (Recommendations for the authors):The mycobacterial cell wall is complex, including an outer mycomembrane, non-covalently attached complex lipids, and depending upon growth conditions, a polysaccharide capsule. Multiple constituents of the cell wall have been shown to have impacts on host cells. In this manuscript, the authors investigate whether different methods used to prepare a single cell suspension of Mtb for infection studies change the mycobacterial cell wall in ways that lead to differential outcomes after infection. To do so, they compared sonication of bacteria followed by a low-speed spin with filtration through a 5uM filter. Sonicating followed by a low-speed spin resulted in an infection that was more inflammatory with higher induction of IL1, IL6, Nos2, and TNF. This increased inflammatory cytokine expression was TLR2 dependent. In addition, they show that sonicated bacteria grow in macrophages whereas filtered bacteria do not. However, the method of preparation of bacteria has no effect in on growth or inflammatory cytokine production in vivo in the mouse model. Finally, they show that mutants lacking the complex lipid PDIM are hyperinflammatory relative to wild type, as has been previously reported, but only when bacteria are prepared by sonication prior to infection. The major strength of this paper is the potential for providing clarity for the entire field around the impact of technical methods for preparing bacteria for infection during macrophage studies, which could bring more consistency and ability to interpret disparate results. The weaknesses are a lack of biological mechanistic insight, the lack of impact on in vivo infections and inconsistencies that make it difficult to draw clear conclusions from the data presented.

We thank the reviewer for their thoughtful assessment. We agree that our findings may help explain inconsistencies in a large body of literature that has been published over decades studying the interaction between *Mycobacterium tuberculosis* (Mtb) and macrophages. Importantly, we hope our manuscript will ensure more robust and reproducible host-pathogen studies in the future. In our revised manuscript, we have tested two additional Mtb strains: HN878, a W-Beijing isolate that was isolated in a TB outbreak in Houston in the 1990s, and Erdman, a strain that is commonly used is laboratory studies. We found that the method of preparation has a similar impact on macrophage inflammatory responses and intracellular bacterial survival for these strains as for H37Rv (added as Figure 2 —figure supplement 1 and Figure 4EF). Thus, our findings are not specific to H37Rv and are likely to impact the field broadly.

In addition, we have revised the manuscript and added analyses to address all of the inconsistencies that were outlined below. We have included a figure that summarizes the data and more clearly integrates the impact of each preparation method into a coherent and mechanistic model (Figure 7). Our electron microscopy studies demonstrate that there is a substantial impact of these procedures on the structure of the mycobacterial cell envelope. The Mtb envelope has been shown by many other groups to contain a variety of pathogen associated molecular patterns (PAMPs) and virulence-associated proteins and lipids. We found that sonication alters the appearance of the envelope and increases macrophage inflammatory responses in a TLR2-dependent manner, consistent with the idea that it exposes these PAMPs. Similarly, we found that filtration removes components of the envelope and also attenuates intracellular survival of the bacilli. Our data support the idea that sonication and filtration are laboratory manipulations that, depending upon the question being investigated, generate experimental artifacts with the potential to lead to misinterpretation. We think that defining the lipid and/or protein changes induced by these laboratory manipulations is complex and beyond the scope of our studies, and, moreover, would not change the importance of our findings.

We initially included the in vivo studies, not because we think sonicated or filtered bacilli are biologically relevant for modeling natural infection, but we were concerned to what degree the artifacts related to preparation methods might also confound the in vivo literature. We found it reassuring that there did not appear to be a substantial impact on early in vivo infection. However, in thinking about the reviewers comments, we appreciate that we cannot exclude an impact of cell preparation on parameters we did not interrogate, such as inflammatory cell recruitment, the adaptive immune response, etc. In addition, we may have missed effects on cytokine responses, since we only measured a limited number of analytes, and we used a total lung sample, rather than interrogating directly infected cells (which would be technically difficult at low dose and early time points). Given these caveats, we have removed the in vivo data from the manuscript. Overall, we agree with the three reviewers that our macrophage findings are rigorous, well substantiated by the data, and have the potential for providing clarity to the field and for bringing more consistency in the future.

1. It is interesting that the authors present data showing a difference in growth in bone-marrow derived macrophages in which sonicated or spun bacteria can grow, but filtered bacteria do no. Many labs do not sonicate bacterial stocks prior to macrophage infection, and even filter the cells, yet do see growth of bacteria in macrophages. In fact, from at least one previous publication from the author's lab, although the methods report that 5uM filtering was used to prepare the bacteria for macrophage infection, they observed growth of the bacteria in macrophages (PMID: 32636249). Can the authors explain this apparent discrepancy?

We appreciate the reviewer’s careful attention to the data. We think that the text of the manuscript and the example that we showed overstated the impact of filtration on intracellular bacterial growth. Across all of the many experiments that we have done, the 55μm-filtered bacteria are impaired relative to non-filtered bacteria, but they do increase in bacterial burden a small amount over time. We have modified the text and replaced the initial figure with more representative data (Figure 4A). In addition, we have added data from HN878 and Erdman strains, which show a similar effect (Figure 4 E and F).

2. There are differences in the way bacteria that have been spun only (sp) behave compared with sonicated and spun (so/sp) vs filtered that make a clear understanding of the impact of these treatments on the bacteria difficult. For example, sp bacteria most closely resemble filtered when comparing inflammatory cytokine gene expression; however like so/sp bacteria they grow in macrophages. Sp bacteria also retain an outer layer (the composition of which is ambiguous) that so/sp bacteria do but do not have membrane blebs. Thus taken together, it is difficult to connect the observed differences in bacterial morphology with the different phenotypes seen during macrophage infection.

We appreciate that the multiple methods of preparation used in this study made it challenging for the reader to conceptualize the impact of each treatment. To address this, we now include a schematic diagram to show how sonicating, centrifuging, and filtering impact cell envelope appearance, macrophage gene expression, and intracellular growth within macrophages (Figure 7). We also now include in the studies bacteria that have been sonicated and filtered (5μmF) to show the impact of sonication (increased inflammatory gene expression) and filtering (reduced viability in macrophages) simultaneously (Figure 2A, C, D, Figure 3A-B, Figure 4A-C, E, F). We made substantial changes to the discussion of the manuscript to more clearly communicate one potential model that explains our data.

3. The authors show data that PDIM mutants are hyper-inflammatory relative to wild-type bacteria, but only if they have been sonicated. This is contradictory to a previous publication that showed that PDIM mutants elicit higher levels of TNFa (also examined in this study) during macrophage infections in which bacteria were not prepared by sonication (PMID: 34755600).

The referenced paper utilized PDIM-deficient ∆*mas* and ∆*ppsD* strains to study the TLR2-dependent host inflammatory response to Mtb. Our data are consistent with the data in that paper, as we have now clarified in the discussion. In that paper, the authors note, “*Tnf* did not cluster with 1B genes; instead *Tnf* was in cluster 3 (Figure 1A) together with genes minimally impacted by PDIM and ESX-1.” When they directly tested the impact of PDIM on *Tnf* expression by qPCR, there were no significant differences noted between H37Rv and the PDIM-deficient strains (see their Figure 3CDE). TNF secretion trended higher in the ∆*ppsD* relative to the WT strain, however this did not reach statistical significance (see their Figure 3G). In the discussion, the authors concluded that “PDIM and ESX-1 only minimally impact expression of the early TLR2-dependent gene cluster, suggesting that the inherent capacity of TLR2 to ‘recognize’ cognate ligand on the bacterium is similar between wild-type Mtb and ESX-1- or PDIM-mutant Mtb.” Overall, ours findings are consistent with theirs, even though there are a variety of differences between their experimental methods and ours (including how BMDMs were differentiated [M-CSF vs. L929 supernatant], time points examined, and MOI [5 vs. 10]).

4. The authors present data showing that there is pro-inflammatory activity contained within the supernatants of bacteria that have been sonicated by applying supernatants from a bacterial suspension that has been passed through a 0.2uM filter to macrophages. However, it is not clear how much of the observed pro-inflammatory activity during infection itself comes from these soluble materials vs the bacteria themselves. If they wash the sonicated bacteria what do they see? It would be nice to see an experiment with so/sp bacteria, sups, and washed so/sp bacteria side by side.

We appreciate the reviewer’s thoughts in this direction. Given that our electron microscopy data showed that the many of the membrane changes in the sonicated samples were associated with the bacterial surface, we did not think washing would effectively release substantial amounts for further study. In addition, if we washed the bacteria after preparing single cells, once we spun them down, they would again aggregate. However, to answer the reviewer’s concern, we have included the data from the so/sp bacteria and the supernatants side-by-side (now included in Figure 2E), which allows the reader to compare the pro-inflammatory activity from the bacilli versus extra-bacterial material. In addition, to better address this question, we exposed BMDMs to the sterile filtrate of so/sp bacteria, spin bacteria, or 5μmF bacteria and found that only the filtrate of the so/sp bacteria increased inflammatory gene expression (now included in Figure 2 —figure supplement 2). Given that the sterile filtrate of the spin bacteria did not induce inflammatory gene expression, we can more confidently attribute these changes to soluble factors following sonication.

Reviewer #2 (Recommendations for the authors):The interaction of Mtb with macrophages is extensively studied by TB investigators and several studies have suggested that the methodology used to prepare Mtb for macrophage infections may influence the macrophage response. In this study the authors aimed to carefully examine this phenomenon with the lab-derived Mtb H37Rv strain. They found that disaggregation methods had a huge impact on macrophage responses in vitro and cautioned that this may lead to experimental artefacts. However, Mtb subjected to different disaggregation methods showed no difference in their in vivo growth and cytokine production in infected mice.Strengths1. The authors tackle an important problem that TB investigators face when performing in vitro macrophage infection studies that may lead to erroneous interpretation of data.2. The data support the conclusions drawn.3. TEM studies to examine damage to bacterial envelop by the two disintegration methods.Weaknesses1. The authors have not considered bacterial cell population heterogeneity.2. Lack of a mechanism for the different in vitro and in vivo outcomes with sonicated and filtered Mtb.3. No data with Mtb clinical strains.

We thank the reviewer for their thoughtful comments. We have now include data for the clinical W-Beijing isolate, HN878, which was recovered from a patient in Houston in the 1990s, as well as another commonly used laboratory strain, Erdman. We had similar findings with H37Rv, HNH878, and Erdman: sonicated bacilli were hyperinflammatory, while those that were filtered were attenuated in macrophages, indicating that our findings extend beyond H37Rv (added as Figure 2 —figure supplement 1 and Figure 4E and F).

In our new manuscript, we have considered bacterial heterogeneity, both incorporating the topic into the discussion and performing additional analysis (added as Figure 2C and Figure 5G). We agree that Mtb cultures are highly heterogenous and that bulk assays, such as intracellular growth, qPCR, and ELISA, fail to take this into account. We considered the possibility that a small minority of the so/sp bacilli are contributing to the heightened inflammatory response, but the Fluoro-dot data, which allows us to visualize at a single cell level, argue against this possibility. If there were a small population of bacilli in the filtered sample that were inhibiting the inflammatory response, then we would have expected that the samples that were mixed would behave like the filtered sample, but rather we saw an intermediate phenotype. Two aspects of heterogeneity that we assessed are clump size and capsule thickness, which are now quantified and included in Figure 2C and 5G, respectively. We found a significant reduction in the thickness of the capsule of the 5μmF bacteria compared to both so/sp and sp bacteria. For all preparation methods, however, there was a wide distribution in cell envelope thickness, which may contribute to heterogeneity in macrophage responses to bacilli on a cell-to-cell level. Finally, we also evaluated the number of bacteria per clump and did not find differences that could explain our findings, so heterogeneity in this aspect of the bacterial populations is unlikely to explain the differences in outcomes.

As we mentioned in the reply to Reviewer 1, we initially included the in vivo studies, not because we think sonicated or filtered bacilli are biologically relevant for modeling natural infection, but we were concerned that the artifacts created by preparation methods might also confound in vivo studies. We found it reassuring that there did not appear to be a substantial impact on early in vivo infection. There are many possible explanations for the in vivo and in vitro differences, including the difference between alveolar macrophages and bone marrow-derived macrophages, the impact of aerosolization, assessments over 7-14 days rather than 3-5 days, etc. However, in thinking about the reviewers’ comments, we appreciate that we cannot exclude an impact of cell preparation on parameters we did not interrogate, such as inflammatory cell recruitment, the adaptive immune response, etc. In addition, we may have missed effects on cytokine responses, since we only measured a limited number of analytes, and we used a total lung sample, rather than interrogating directly infected cells (which would be technically difficult at low dose and early time points). Given these caveats with the in vivo data, we have removed them from the manuscript.

In this study, the authors examined the impact of two routinely used bacterial disaggregation techniques on the subsequent interaction of Mtb with macrophages. They found that sonicated bacilli induced a strong hyperinflammatory response in BMDMs and still replicated normally in the macrophages. In stark contrast, Mtb that had been passed through a filter induced little inflammatory response in BMDMs and were also highly growth attenuated. Absence of PDIM impacted sonicated but not filtered bacteria. Interestingly, disaggregation methods did not impact growth in AMs or in vivo. In conclusion, the data presented here indicate that macrophage responses are strongly affected by the bacterial disaggregation methodology used and this may confound the results of Mtb-macrophage interaction studies.Comments1. The authors should rule out the possibility that their findings could be biased by cell population heterogeneity in Mtb H37Rv.

We have included discussion about the potential impact of heterogeneity, and to assess a potential impact of bacterial heterogeneity in our results, we further analyzed microscopy images to quantify variation in bacterial envelope thickness and used immunofluorescence to quantify clump size for each preparation method. When we evaluated capsule thickness, there was significant reduction in the thickness of the bacteria in the 5μmF samples compared to both so/sp and sp samples, but in all cases, there is a wide distribution, which may contribute to heterogeneity in macrophage responses on a cell-to-cell level. We considered the possibility that a small minority of the so/sp bacilli are contributing to the heightened inflammatory response, but the fluoro-dot data (which provided single cell resolution of TNF secretion) argue against this possibility. We also evaluated the number of bacteria per clump and filtered (5μmF and so/5μmF) preparations had slightly more single/doublets and slightly fewer clumps than the other samples (so/sp and sp; Figure 2C). But the aggregation status of different preparation did not explain the differences that we see in macrophage responses.

2. A lipid analysis of the bacteria after sonication and filtration would help substantiate the authors' conclusions.

Our electron microscopy studies demonstrate that there is an obvious structural impact of sonication and filtration on the mycobacterial cell envelope. We attempted to further characterize the differences in lipid composition, in collaboration with an expert in Mtb lipidomics (Fong-Fu Hsu, Washington University School of Medicine). It is standard in Mtb lipidomic studies to grow the bacilli on plates to eliminate need for detergents, which interfere with the mass spectrometry analysis. However, to compare the bacterial preparation methods described here requires adapting the analysis to liquid cultures. We were unable to establish conditions in which we could both reliably quantify and normalize across samples while also minimizing or removing the detergent so as to not interfere with the analysis. In the end, while we could see that 5umF bacilli lacked numerous envelope lipids, we were unable to obtain data that we consider publishable. We think that to rigorously define the lipid, protein and/or architectural changes induced by these laboratory manipulations is beyond the scope of these studies, and, importantly, would not change the importance and impact of our findings.

3. Data from additional Mtb strains is required to strengthen the observations.

To extend our findings to other laboratory strains and clinical strains, we measured intracellular growth of Mtb in BMDMs and BMDM gene expression using Erdman and HN878 strains alongside H37Rv and obtained comparable results. As in H37Rv, these strains elicited increased inflammatory gene expression in BMDMs when sonicated (Figure 2 —figure supplement 1). Further, the filtered bacteria were growth restricted in BMDMs (Figure 4E and F).

Reviewer #3 (Recommendations for the authors):In this work, Mittal et al. determine how three commonly used methods for preparing Mtb for use in cellular and animal models of infection impact host response and infection outcomes. While largely technical, the key findings of the manuscript are important for understanding the existing literature of TB infection biology, and will undoubtedly inform how groups studying Mtb virulence in cellular and animal models perform and interpret their experiments in the future.The primary comparisons were between sonication, filtration through a 5um filter, and centrifugation. Key findings:1) Macrophages responded markedly differently to Mtb that were sonicated, filtered to make single-cell suspensions, or centrifuged- the primary initial outcome evaluated was comprehensive transcriptional profiling at 72 hours.2) Mtb preparation using sonication results in enhanced inflammatory responses relative to uninfected or filtered cells. Outcomes evaluated include focused inflammatory gene transcriptional responses at 6 hours and 24 hours and TNF secretion as measured by FluoroDOT assays and TNF secretion and IL-1b secretion as measured by ELISA. Further, this enhanced inflammation is conferred by the sterile filtrate of the sonicated Mtb preparation, suggesting shedding of bacterial components into the media.3) Mtb preparation using filtration results in reduced growth in bone marrow-derived macrophages but no impaired growth in alveolar macrophages or in vivo in mice.4) TEM and uranyl acetate staining showed structural differences in the outer layers of Mtb prepared using sonication, filtration, and centrifugation. Sonicated Mtb had associated extra-bacterial vesicles; filtered Mtb had a thin outer coat suggesting loss of the outer membrane and capsule.5) The impact of PDIM on early macrophage inflammatory responses as measured by targeted inflammatory gene expression and TNF FluorDOT assay was notable only for sonicated Mtb.The finding that the preparation of Mtb changes inflammatory responses is compelling and well-supported by the transcriptional and cytokine data, in particular for the enhanced response to sonicated bacteria. The findings that this enhanced inflammation is partially TLR2 dependent and is associated with a soluble component of the bacterial preparation are similarly solid.The finding that filtration results in reduced Mtb growth in BMDM relative to sonicated or centrifuged Mtb would be stronger if macrophage survival was assayed in parallel. At the MOIs used, Mtb infection is well-described to result in some degree of macrophage cell death, which can significantly impact recovery of Mtb from macrophage cell cultures. Clumping has also been described to have implications for macrophage cell death; given that the authors show different levels of clumping for bacteria prepared using each method, this is a particularly important outcome to measure in parallel with CFU.The interpretation that the TEM findings and uranyl acetate staining (small extra bacterial vesicles for sonicated Mtb and loss of membrane/capsule for filtered Mtb) likely reflect the underlying reasons for differences in macrophage responses to the bacteria is consistent with the data, but not explicitly tested (and not readily testable).The interpretation that differences in pre-existing macrophage inflammatory state in BMDM vs. alveolar macrophages drive the differences observed in the intracellular growth of filtered Mtb is also consistent with the data but not explicitly tested.The overall conclusion that the specific methods used to prepare Mtb for infection impact macrophage responses and infection outcomes is solid and supported by the data. The conclusion that the method of bacterial preparation should be considered in both interpreting existing literature and experimental planning is also well-founded based on the data shown.

We thank the reviewer for the detailed summary of our findings. We agree that bacterial clumping likely has significant implications on macrophage-Mtb interactions, including cell death. To better evaluate this, we assessed macrophage cell death at days 0, 3, and 5 post-infection, alongside our intracellular bacterial survival assay. We did not find substantial differences based upon preparation method (now included as Figure 4C) that could explain our findings. We also utilized the microscopy images to evaluate heterogeneity in clump size between different preparations. The majority of the bacilli in all preparations were single or double, but there was considerable heterogeneity within each preparation, and there were more small clumps in the so/sp and sp samples. Since both so/sp and sp had similar distributions, clumpiness is unlikely to explain the macrophage inflammatory differences. We appreciate the reviewer’s comments regarding ex vivo infection of alveolar macrophages and in vivo infection of mice. On further review, our hypotheses were not well supported by the data. Along with the concerns of the other reviewers, we have removed the in vivo and alveolar macrophage data. We think that further phenotypic characterization of macrophage populations, while potentially interesting, would not change the conclusions of our manuscript or the impact of our study.

1) No testing of macrophage viability: Macrophage viability is another critical virulence related infection outcome, and can confound CFU measurements- in particular at higher MOIs and later timepoints. Measuring macrophage viability in parallel with CFU measurements would provide an important control for the CFU data and would offer insight into how the method of Mtb preparation impacts this additional infection outcome metric.

We used an ATP-based luminescence assay (CellTiterGlo, Promega) to evaluate viability of BMDMs infected at a multiplicity of infection (MOI) of 10 at 0, 3, and 5 days post-infection, alongside a replicate of our intracellular bacterial survival assay. There was a significant, but modest, increase in cell death in the spun bacilli at 5 days post-infection compared to uninfected macrophages and all other infected samples (perhaps related to a small number of large clumps in this preparation [see Figure 2C]). The cell death data has been added as Figure 4C. Overall, we conclude that the differences in intracellular growth were not explained by differences in macrophage viability (Figure 4C).

2) No testing of bacterial viability: While the relatively similar day 0 CFU in the macrophage infection assays suggest that similar numbers of live bacteria come out of each preparation method, the data would be stronger if bacterial viability were explicitly tested immediately after each preparation (using a live-dead reporter strain, or even CFU of the in vitro bacteria).

In addition to plating bacteria four hours after the macrophage infections, we verified that each preparation of Mtb used for the infection (input) produced roughly equivalent viable colonies when cultured on 7H10 plates. We found that the conversion between optical density and viable colony counts was slightly different for the filtered (5umF and so/5umF) bacilli compared to the other preparation (1 OD_600_ = 3 x 10^8^ bacteria per mL for all preparation except filtered, which was 1 OD_600_ = 2 x 10^8^ bacteria per mL). On the basis of viable colonies obtained on 7H10 plates, we calculated the MOI in each experiment. This has now been clarified in the methods. Furthermore, we freshly prepared so/sp, sp, or 5μmF and grew them in 7H9 liquid culture and confirmed that their growth patterns were indistinguishable from one another (Figure 4D).

3) The hypothesis that the baseline inflammatory state of the macrophage influences whether filtered Mtb can grow is testable and potentially worth testing to more definitively support this idea. This could be done by pre-stimulating the macrophages with a range of that change the inflammatory and/or polarization state of the cells, then infecting with Mtb prepared using the three methods profiled.

In our previous submission, we discussed differences between the response of BMDMs and alveolar macrophages. On further review, the conclusions that we drew were overly speculative. To address this, we have removed data related to ex vivo infection of alveolar macrophages, as well as the in vivo infections. While further testing of a variety of macrophages is possible, the preparation techniques we describe do not have clinical correlates to human TB disease, and we feel that this is beyond the scope of our paper. Recognizing that research groups use a variety of primary cells and often polarize cells before or after infection with Mtb, we recommend that they validate that their findings are not impacted by their method of bacterial preparation (or at least report their methodology and consider avoiding sonication and filtration).